# ANDROIDWORLD: A DYNAMIC BENCHMARKING ENVIRONMENT FOR AUTONOMOUS AGENTS

Christopher Rawles[*1], Sarah Clinckemaillie[†2], Yifan Chang[†2], Jonathan Waltz[2], Gabrielle Lau[2], Marybeth Fair[2], Alice Li[1], William Bishop[1], Wei Li[1], Folawiyo Campbell-Ajala[1], Daniel Toyama[1], Robert Berry[1], Divya Tyamagundlu[2], Timothy Lillicrap[1], and Oriana Riva[1]

[1]*Google DeepMind*
[2]*Google*

## ABSTRACT

Autonomous agents that execute human tasks by controlling computers can enhance human productivity and application accessibility. However, progress in this field will be driven by realistic and reproducible benchmarks. We present ANDROIDWORLD, a fully functional Android environment that provides reward signals for 116 programmatic tasks across 20 real-world Android apps. Unlike existing interactive environments, which provide a static test set, ANDROIDWORLD dynamically constructs tasks that are parameterized and expressed in natural language in unlimited ways, thus enabling testing on a much larger and more realistic suite of tasks. To ensure reproducibility, each task includes dedicated initialization, success-checking, and tear-down logic, which modifies and inspects the device's system state.

We experiment with baseline agents to test ANDROIDWORLD and provide initial results on the benchmark. Our best agent can complete 30.6% of ANDROIDWORLD's tasks, leaving ample room for future work. Furthermore, we adapt a popular desktop web agent to work on Android, which we find to be less effective on mobile, suggesting future research is needed to achieve universal, cross-platform agents. Finally, we also conduct a robustness analysis, showing that task variations can significantly affect agent performance, demonstrating that without such testing, agent performance metrics may not fully reflect practical challenges. ANDROIDWORLD and the experiments in this paper are available at https://github.com/google-research/android_world.

## 1 INTRODUCTION

Autonomous agents that interpret natural language instructions and operate computing devices can provide enormous value to users by automating repetitive tasks, augmenting human intelligence, and accomplishing complex workflows. However, a key research challenge remains the realistic evaluation of these agents in real-world settings. Despite growing enthusiasm for building autonomous agents (Deng et al., 2023; Rawles et al., 2023; Zheng et al., 2024a; Koh et al., 2024; Kim et al., 2024; He et al., 2024; Gravitas, 2023; Wu et al., 2023; Xie et al., 2023) most existing approaches for evaluation compare an agent's actions at each step to a previously collected human demonstration (Deng et al., 2023; Rawles et al., 2023; Yang et al., 2023b; Zhang & Zhang, 2023; Lù et al., 2024; Zhang et al., 2024c; Yan et al., 2023; Li et al., 2024). Measuring performance in this way can be misleading because when performing tasks online in real environments agents can take multiple paths to solve tasks, environments may behave non-deterministically, and agents can dynamically learn from mistakes to correct their actions (Shinn et al., 2023; Liu et al., 2018b; Li et al., 2023b; Pan et al., 2024). For this reason, online evaluation of agents in realistic environments able to reward task outcome provides a gold standard for evaluation. While there is an emerging body of work to address this need across different environments (Zhou et al., 2023; Koh et al., 2024; Drouin et al.,

---

[*]Lead contributor. Contact: `crawles@google.com`
[†]Equal contribution.

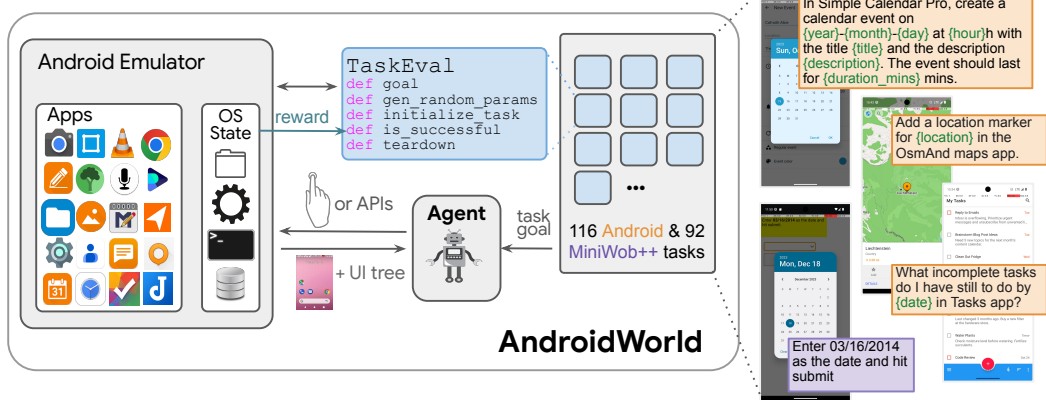

Figure 1: ANDROIDWORLD is an environment for building and testing autonomous agents.

2024; Lee et al., 2024; Xie et al., 2024; Bonatti et al., 2024; Zheng et al., 2024b), there is no comprehensive solution for mobile platforms, such as Android, which are used by billions of users and therefore represent environments in which automation agents may be very productively employed. We introduce ANDROIDWORLD to address this.

At its core, ANDROIDWORLD offers a reliable means of obtaining reward signals for tasks performed by agents in realistic mobile environments. Reward signals are quantitative metrics that indicate functional correctness of a task, i.e. is the stated goal achieved? For example, for the task "Send a text message to Jane confirming I'll be there," a positive reward indicates that the relevant message has been sent. Unlike simulated environments (Tassa et al., 2018; Shridhar et al., 2020) or games (Mnih et al., 2013; Silver et al., 2016; Vinyals et al., 2019; Wang et al., 2023b; Tan et al., 2024; Toyama et al., 2021), real-world apps and websites do not inherently offer explicit reward signals. While human (Rawles et al., 2023; Zheng et al., 2024a; Pan et al., 2024; Kinniment et al., 2023) or LLM-based (Chiang et al., 2024; Zheng et al., 2023; Liu et al., 2023; Du et al., 2023; Ma et al., 2023; Pan et al., 2024; He et al., 2024) judges can be employed to reward the outcome of a task, these approaches scale poorly or are not fully reliable, respectively. Alternatively, environments for autonomous agents which provide automated ground-truth rewards for complex workflows have been developed (Yao et al., 2023; Zhou et al., 2023; Koh et al., 2024; Xie et al., 2024; Bonatti et al., 2024). We find two problems with these environments. First, they are constrained to desktop computing environments, overlooking the mobile domain, which is of paramount importance given the ubiquity and diversity of mobile devices in the real world. Secondly, they are limited in their real-world diversity and scale. Crucially, unlike in real-world scenarios where conditions and task inputs vary widely, these environments support only static test specifications, meaning that when task parameters deviate, the reward signal is likely to break.

We seek to develop a comprehensive benchmark that addresses the limitations of the existing approaches above for evaluating automation agents in mobile environments. ANDROIDWORLD does this by spanning 20 Android apps on a total of 116 programmatic tasks to provide ground truth-rewards. Unlike existing test environments (MiniWoB++ (Shi et al., 2017) being a notable exception), each task in ANDROIDWORLD is dynamically instantiated using randomly-generated parameters, challenging agents with millions of unique task goals and conditions. While MiniWob++ consists of simple, synthetic websites, ANDROIDWORLD leverages actual Android applications. A main challenge that ANDROIDWORLD must address is how to ensure that reward signals are durable when using real-world applications and varying task parameters dynamically. ANDROIDWORLD's solves this by leveraging the extensive and consistent state management capabilities of the Android OS, using the same mechanisms that the apps themselves utilize to store and update data.

In addition to providing a comprehensive benchmark, ANDROIDWORLD is lightweight, requiring only 2 GB of memory and 8 GB of disk space, and is designed with convenience in mind. It connects agents to the Android OS by leveraging the Python library AndroidEnv (Toyama et al.,

Table 1: Comparison of different datasets and environments for benchmarking computer agents.

| | Env? | # of apps or websites | # task templates | Avg # task instances | Reward method | Platform |
|---|---|---|---|---|---|---|
| GAIA | ✗ | n/a | 466 | 1 | text-match | None |
| MIND2WEB | ✗ | 137 | 2350 | 1 | None | Desktop Web |
| WEBLINX | ✗ | 155 | 2337 | 1 | None | Desktop Web |
| WEBVOYAGER | ✗ | 15 | 643 | 1 | LLM judge | Desktop Web |
| PIXELHELP | ✗ | 4 | 187 | 1 | None | Android |
| METAGUI | ✗ | 6 | 1125 | 1 | None | Android |
| MOTIF | ✗ | 125 | 4707 | 1 | None | Android (Apps+Web) |
| AITW | ✗ | 357+ | 30378 | 1 | None | Android (Apps+Web) |
| ANDROIDCONTROL | ✗ | 833 | 15283 | 1 | None | Android (Apps+Web) |
| OMNIACT | ✗ | 60+ | 9802 | 1 | None | Desktop (Apps+Web) |
| ANDROIDARENA | ✗ | 13 | 221 | 1 | Action match/LLM | Android (Apps+Web) |
| LLAMATOUCH | ✗ | 57 | 496 | 1 | Screen match | Android (Apps+Web) |
| MINIWOB++ | ✓ | 1 | 114 | ∞ | HTML/JS state | Web (synthetic) |
| WEBSHOP | ✓ | 1 | 12k | 1 | product attrs match | Desktop Web |
| WEBARENA | ✓ | 6 | 241 | 3.3 | url/text-match | Desktop Web |
| VISUALWEBARENA | ✓ | 4 | 314 | 2.9 | url/text/image-match | Desktop Web |
| WORKARENA | ✓ | 1 | 29 | 622.4 | cloud state | Desktop Web |
| MOBILE-ENV | ✓ | 1 | 13 | 11.5 | regex | Android (Apps) |
| B-MOCA | ✓ | 4 | 6 | 1.9 | regex | Android (Apps+Web) |
| MMINA | ✓ | 14 | 1050 | 1 | text-match | Desktop web |
| OSWORLD | ✓ | 9 | 369 | 1 | device/cloud state | Desktop (Apps+Web) |
| WINDOWSAGENTARENA | ✓ | 11 | 154 | 1 | device state | Desktop (Apps+Web) |
| AGENTSTUDIO | ✓ | 9 | 205 | 1 | device state | Desktop (Apps+Web) |
| **ANDROIDWORLD** | ✓ | 20 | 116 | ∞ | device state | Android (Apps+Web) |

2021) to connect to the freely available Android Emulator.[1] In addition to the 116 Android tasks, we extend ANDROIDWORLD with web tasks by integrating the MiniWoB++ (Shi et al., 2017; Liu et al., 2018a) benchmark into it.

To demonstrate ANDROIDWORLD's usefulness as a benchmark, we build and release a multi-modal agent, M3A (Multimodal Autonomous Agent for Android), and establish state-of-the-art results on ANDROIDWORLD. We analyze M3A's performance using both multimodal and text-only input, and we observe that while multimodal perception can improve performance in some cases, it generally does not outperform the text-only approach. On ANDROIDWORLD, M3A achieves a 30.6% success rate, which surpasses that of a web agent adapted for Android but remains significantly lower than the human success rate of 80.0%. In pursuit of building robust UI control agents, our study includes comprehensive tests under varied real-world conditions, demonstrating significant performance variations primarily driven by changes in intent parameters.

We make the following contributions: (i) the creation of a new, highly diverse and realistic mobile UI control agent environment; (ii) establishment of benchmark performance with a state-of-the-art multimodal agent, and (iii) a careful analysis demonstrating the need to evaluate agents across variable task parameters and conditions due to the inherent stochasticity in both models and environments.

## 2 RELATED WORK

Table 1 compares existing evaluation environments for autonomous UI agents.

### 2.1 INTERACTIVE EVALUATION ENVIRONMENTS

Effective evaluation of autonomous agents requires benchmarks that mimic real-world scenarios, but also interactive environments that provide reward signals upon successful task completion (Rawles et al., 2023; Deng et al., 2023; Abramson et al., 2022; Ruan et al., 2023; Chen et al., 2021). Many existing benchmarking environments target web browsing. MiniWoB++ (Shi et al., 2017; Liu et al., 2018b) consists of small, synthetic HTML pages with parameterizable tasks which allow for un-

---

[1]The Android Emulator is packaged as part of Android Studio, which can be downloaded from https://developer.android.com/studio

limited task variability. WebShop (Yao et al., 2023) provides a simulated e-commerce environment, whereas WebArena (Zhou et al., 2023) and VisualWebArena (Koh et al., 2024) consist of simulated websites across up to six domains. WorkArena (Drouin et al., 2024) consists of 29 tasks for enterprise software. GAIA (Mialon et al., 2023) is a static dataset that tests an agent's ability to interact with live web environments. MMInA (Zhang et al., 2024e) is a multihop and multimodal benchmark designed to evaluate agents for compositional Internet tasks.

Towards building computer use agents, OSWorld (Xie et al., 2024), WindowsAgentArena (Bonatti et al., 2024), and AgentStudio (Zheng et al., 2024b) provide a test suite of tasks for desktop computer interfaces and custom execution-based evaluation scripts across 9, 11, and 9 apps, respectively. In the mobile domain, existing benchmarks are limited and do not capture the diversity of real-world mobile interactions, containing low-complexity tasks or on a limited number of applications. B-MoCA's (Lee et al., 2024) evaluation is based on 6 simple tasks (e.g., "Call 911", "turn on airplane mode") across 4 apps[2], validated using regular expressions. Mobile-Env (Zhang et al., 2024b) offers task reproducibility limited to 13 task templates for a single app (WikiHow).

While ANDROIDWORLD shares the mobile OS focus of B-MoCA and Mobile-Env, it is more comparable to OSWorld (and WindowsAgentArena, which builds on top of OSWorld) in terms of task complexity and the diversity of interactions it supports. ANDROIDWORLD enhances OSWorld's approach by dynamically constructing the start states of an agent's run and varying the task parameters in unlimited ways, thus allowing for a new type of evaluation under varying real-world conditions.

Other studies leverage human evaluation (Rawles et al., 2023; Zheng et al., 2024a; Bishop et al., 2024) for tasks where automatic evaluation is not available. Lastly, emerging research (Pan et al., 2024; He et al., 2024; Xing et al., 2024; Zheng et al., 2024b) explores the potential of multimodal models to generalize agent evaluations to new settings, though this area requires further research to achieve accuracy comparable to manually-coded rewards.

AndroidEnv (Toyama et al., 2021) provides a mechanism to manage communication with the Android emulator, similar to Playwright and Selenium for web environments. While ANDROIDWORLD leverages this functionality, it diverges in its reward system. AndroidEnv's approach requires modifying application source code and implementing task-specific logging statements, making it well-suited for gaming environments with easily verifiable success criteria. In contrast, ANDROID-WORLD implements a non-invasive reward mechanism, allowing it to create a benchmark suite for apps whose source code is unavailable and to reuse validation components across different apps. This approach enables ANDROIDWORLD to cover a broader range of real-world mobile tasks.

## 2.2 STATIC DATASETS FOR UI AUTOMATION

Datasets derived from human interactions provide proxy metrics that correlate with real-world agent performance (Li et al., 2020; Burns et al., 2021; Deng et al., 2023; Rawles et al., 2023). On mobile platforms, AitW (Rawles et al., 2023), AndroidControl (Li et al., 2024), PixelHelp (Li et al., 2020), AndroidArena (Xing et al., 2024), LlamaTouch (Zhang et al., 2024d), UGIF (Venkatesh et al., 2022), and MoTIF (Burns et al., 2021) consist of demonstrations across Android apps and mobile websites, with screens often represented via accessibility trees. In contrast, desktop web environments typically utilize the DOM for representing website content, with Mind2Web (Deng et al., 2023), OmniAct (Kapoor et al., 2024) and others, across various desktop websites. Mobile-based datasets frequently involve more complex actions, such as scrolling, which are not as useful in DOM-based desktop interactions where the entire action space is readily accessible. Additionally, API-centric datasets like API-Bank (Li et al., 2023a), ToolTalk (Farn & Shin, 2023), and ToolBench (Xu et al., 2023) assess agents' capabilities to manipulate computer systems via APIs.

## 2.3 INTERACTIVE AGENTS

Prior to today's foundation models, traditional approaches to developing user interface-operating agents primarily used reinforcement learning and behavioral cloning to simulate interactions like mouse clicks and keyboard typing (Liu et al., 2018b; Li et al., 2020; Shvo et al., 2021; Gur et al., 2022a; Humphreys et al., 2022). More recent work leverages off-the-shelf foundational models (Gemini, 2023; OpenAI, 2023; Touvron et al., 2023) with in-context learning (ICL) and fine-tuning

---

[2]Based on what reported in the Experiments Section of the B-MoCA manuscript as of October 1[st], 2024.

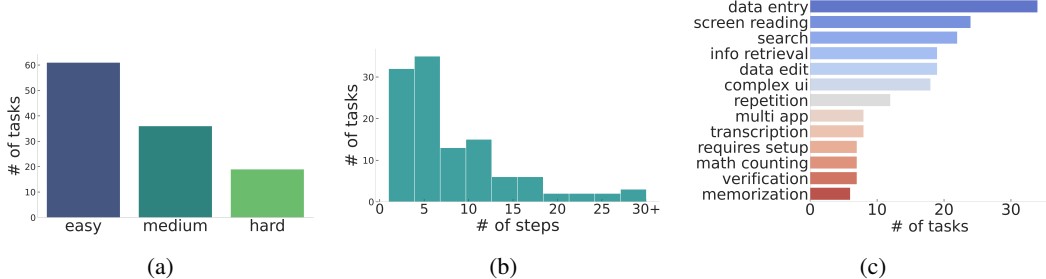

Figure 2: Annotators performed the tasks assigned to them, assigned a difficulty level (2a) and estimated the number of steps required to complete each task (2b), using the action space available to an agent. For each task, they selected relevant category tags from a predefined list (2c).

applied to mobile (Rawles et al., 2023; Hong et al., 2023; Wang et al., 2023a; Yan et al., 2023; Zhang & Zhang, 2023; Bishop et al., 2024; Zhang et al., 2023), desktop web (Zheng et al., 2024a; Deng et al., 2023; Zhou et al., 2023; Koh et al., 2024; Cheng et al., 2024; Lai et al., 2024; You et al., 2024), and desktop OS (Wu et al., 2024; Zhang et al., 2024a; Xie et al., 2024). Recent work explores agents that reflect on system state (Shinn et al., 2023; Yao et al., 2022; Madaan et al., 2024) by leveraging exploration, self-evaluation, and retry-capabilities for continual learning and adaptation (Li et al., 2023b; Yang et al., 2023b; Pan et al., 2024; Wu et al., 2024; Gao et al., 2023; Murty et al., 2024).

# 3 ANDROIDWORLD

## 3.1 ANDROID FOR AUTONOMOUS AGENTS

Android is an ideal environment for developing autonomous agents. It is the most widely-used OS globally[3] and is highly flexible for research, while providing an open world of the Web[4] and over 2M apps for agents to operate in. Using emulation, an Android environment is easy to deploy, does not require specialized hardware, and can be run on a laptop. Android Virtual Devices or emulator images are well suited for research as they are self-contained, easy to distribute, and configurable.

Compared to desktops, mobile environments like Android present unique challenges for computer-use agents. While mobile UIs are simpler due to smaller screens, their action space is more complex, requiring intricate gestures (e.g., navigating carousels, long-pressing, multi-finger zooming) and often more steps to complete tasks. Unlike web-browser-only environments, Android, as an OS, offers greater flexibility, including function-calling APIs (e.g., sending texts) alongside standard UI actions (click, scroll, type).

## 3.2 THE OBSERVATION AND ACTION SPACE

ANDROIDWORLD provides an interface for agents to receive observations and execute actions on Android. It uses AndroidEnv (Toyama et al., 2021) and the Android Device Bridge to facilitate interaction between Android and the agent. The observation space consists of a full-resolution screenshot and a UI tree representation developed for accessibility purposes. The action space is similar to that which humans use, consisting of gestures (i.e., tapping, swiping), typing, and navigation buttons (i.e., go home and go back). In addition to these naturalistic actions, ANDROIDWORLD exposes a limited set of function calling APIs, such as send_text_message, to help agents accomplish goals. Appendix C provides more details on the observation format and action space.

## 3.3 REPRODUCIBLE AND PARAMETERIZED TASKS

ANDROIDWORLD consists of a suite of 116 tasks, spread across 20 diverse applications (see Appendix D for more details). These tasks simulate practical, everyday activities, including note-

---

[3] https://gs.statcounter.com/os-market-share
[4] Mobile is the most popular platform for accessing the web; https://gs.statcounter.com/platform-market-share/desktop-mobile/worldwide/

Table 2: Selected tasks with code describing validation logic.

| Task | Validation code |
|---|---|
| In Simple Calendar Pro, create a calendar event on {event.year}-{event.month}-{event.day} at {event.hour}h with the title '{event.title}' and the description '{event.description}'. The event should last for {event.duration} mins. | `event_exists(event)` |
| Send a text message to {phone_number} with message: {message}. | `message_exists(phone_number, message, messaging_db)` |
| Create a new drawing in Simple Draw Pro. Name it {file_name}. Save it in the Pictures folder. | `file_exists(file_path)` |
| Create a timer with {hours} hours, {minutes} minutes, and {seconds} seconds. Do not start the timer. | `timer_displays(time, ui_hierarchy)` |
| Create a new note in Markor named {file_name} with the following text: {text}. Share the entire content of the note with the phone number {number} via SMS. | `(file_exists(file_name, content=text) + message_exists(phone_number, message)) / 2.0` |
| Turn on WiFi and open {app_name}. | `(wifi_enabled() + app_launched(app_name))/2.0` |

taking, scheduling appointments, communicating through messaging, and interacting with system utilities. The suite consists of open-source apps and built-in Android system apps, such as Settings and Contacts. As rated by humans, the tasks vary in difficulty, duration, and categories (Figure 2).

To achieve a high degree of reproducibility in real-world scenarios, ANDROIDWORLD precisely controls the OS and app states in several ways. The Android OS is fixed, consisting of a Pixel 6 emulator running Android 13. At the start of each task, ANDROIDWORLD resets the device times-tamp to October 15th, 2023 at 15:34 UTC, ensuring consistent time-dependent behaviors across all executions. All applications in ANDROIDWORLD are fully-functional and consists of both open-source apps and OS-level apps included with Android. For the open-source apps, ANDROIDWORLD maintains a constant environment by installing a fixed version of each app, acquired from F-Droid.[5] OS-level apps' versions are determined by the Android OS, which is also fixed. To maintain a reproducible environment, ANDROIDWORLD utilizes apps that do not require login/authentication and can store their application data on device.

In addition to managing the states of apps and operating systems, ANDROIDWORLD precisely defines and controls the state during task execution. Each task has its own unique setup, reward determination logic, and teardown procedures (see Appendix D.2 and D.3 for more details), ensuring a fully reproducible suite of tasks.

Automatic task *parameterization* is a critical mechanism, unique to ANDROIDWORLD, to evaluate agents on a much larger and more realistic suite of tasks than current benchmarks support. Achieving this requires significantly more effort than randomly generating new task parameters because it involves developing evaluation logic that remains valid across different task instantiations. It is exactly through its careful state management that in addition to reproducibility AndroidWorld ensures that the reward mechanisms function correctly. Task parameters, initialized randomly at the start of each task based on a controlled random seed, dictate the initial state and influence reward outcomes. Similar to MiniWoB++ (Shi et al., 2017; Liu et al., 2018a), ANDROIDWORLD consists of a practically infinite set of varying initial conditions and success criteria.

This approach enables finer-grained analyses of agent adaptability, essential for real-world deployment. Beyond robustness testing, dynamic task construction supports online learning, particularly reinforcement learning (Shi et al., 2017; Liu et al., 2018a; Humphreys et al., 2022; Gur et al., 2022a), while also streamlining train/test dataset generation for supervised learning (Humphreys et al., 2022; Shaw et al., 2023; Furuta et al., 2023).

## 3.4 DURABLE REWARDS FROM SYSTEM STATE

ANDROIDWORLD provides reward signals primarily by managing application state using the Android Debug Bridge (`adb`), while also incorporating UI element validation where appropriate. With

---

[5]https://f-droid.org/

`adb`, ANDROIDWORLD has complete access to system resources including the file system, application databases, and system settings. For tasks where system state inspection is impractical, ANDROIDWORLD validates task completion by examining UI elements on screen. Determining reward signals from system state has several benefits. It is highly accurate because an application's state can be quickly inspected and manipulated using the same mechanisms that the app itself utilizes. Using the underlying system state is much more durable than matching superficial UI changes. Additionally, it facilitates easy re-use across disparate apps, which tend to use the same underlying caching mechanisms. For instance, logic for checking existence of a specific file is used across many unrelated applications, including those for file management, note-taking, and media playback. For applications leveraging SQLite databases, a common pattern, ANDROIDWORLD implements evaluators that verify the existence of new and deleted rows. Table 2 shows examples of the validators in ANDROIDWORLD. See Table 6 for a comprehensive list of all tasks in the suite. Table 5 provides selected examples with additional implementation details.

## 3.5 TASK COMPOSABILITY

Inferring task success from system state enables accurate, reusable evaluations and simplifies creating *composite* tasks by combining existing ones. For instance, "Create a calendar event with details and text the details to contact" merges two standalone tasks, facilitated by hermetic initialization and success detection. Composite tasks are more challenging due to their complexity but provide partial rewards for subtask completion, aiding hill climbing. The last two rows of Table 2 show validation code for composite tasks.

## 3.6 INTEGRATING MINIWOB++

We implement MiniWoB++ in the ANDROIDWORLD framework and term it MobileMiniWoB++. Each MobileMiniWoB++ task is instantiated using the standard ANDROIDWORLD interface, inheriting from `TaskEval` base class, and contains methods like `initialize_state` and `is_successful`. Since MiniWoB++ leverages JavaScript for task configuration and success detection, we built a WebView app to communicate between Python and the app.

MobileMiniWoB++ introduces modifications in both observations and actions compared to the original benchmark. For example, HTML5 <input> elements are rendered with native Android UI widgets like the date-picker (see Figure 4), enhancing the realism of the tasks. MobileMiniWoB++ uses the same observation space as the Android tasks (accessibility tree and screenshot). Notably, it does not include the DOM as in the original implementation. The action space from ANDROIDWORLD is retained. We manually review and test each task to ensure they are solvable. We excluded twelve of the original tasks that failed to render correctly on Android, presented compatibility issues with the touch interface, or required near real-time interaction, which poses challenges on emulators. Overall, ANDROIDWORLD supports 92 MiniWoB++ tasks. See Appendix C.3 for more details.

## 4 ANDROIDWORLD AS A COMPUTER-CONTROL BENCHMARK

To test ANDROIDWORLD's applicability for autonomous agents, we develop and test a state-of-the-art agent and its variants across all 20 apps and 116 tasks, as well as on MobileMiniWoB++.

### 4.1 COMPUTER USE AGENTS

#### 4.1.1 M3A

We develop a multimodal autonomous agent for Android, M3A. It is zero-shot, integrating ReAct-style (Yao et al., 2022) and Reflexion-style (Shinn et al., 2023) prompting to consume user instructions and screen content, reason, take actions, and update its decision-making based on the outcome of its actions.

In the first stage, M3A generates an action, represented in JSON, and reasoning for that action. To generate this output, the agent is provided with a list of available action types, guidelines for operating the phone, and a list of UI elements derived from the Android accessibility tree's leaf nodes. The agent receives the current screenshot and a Set-of-Mark (SoM) (Yang et al., 2023a)

Table 3: Success Rates (SR) on ANDROIDWORLD and MobileMiniWoB++.

| Agent | Input | Base model | SR$_{\text{ANDROIDWORLD}}$ | SR$_{\text{MobileMiniWoB++}}$ |
|---|---|---|---|---|
| *Human* | *screen* | *N/A* | *80.0* | *100.0* |
| SeeAct (Zheng et al., 2024a) | SoM (screen + a11y tree) | GPT-4 Turbo | 15.5 | 66.1 |
| M3A-Simple | a11y tree | Gemma 2 | 3.4 | 35.5 |
| M3A-Simple | a11y tree | Gemini 1.5 Pro | 14.7 | 55.2 |
| M3A-Simple | a11y tree | GPT-4 Turbo | 19.8 | **67.7** |
| M3A | a11y tree | Gemma 2 | 9.5 | 45.6 |
| M3A | a11y tree | Gemini 1.5 Pro | 19.4 | 57.4 |
| M3A | SoM (screen + a11y tree) | Gemini 1.5 Pro | 22.8 | 40.3 |
| M3A | a11y tree | GPT-4 Turbo | **30.6** | 59.7 |
| M3A | SoM (screen + a11y tree) | GPT-4 Turbo | 25.4 | **67.7** |

annotated screenshot, which includes bounding boxes with numeric labels on the top-left corner for each UI element (see screenshot in Figure 5). The agent attempts to execute outputted action by referencing the specific mark (if applicable). In addition to the multimodal agent, we have developed a text-only variant that consumes the screen represented using the accessibility tree and selects the relevant action in JSON format.

After executing an action, M3A reflects on its effect by observing any state changes that may have occurred. During this stage, the agent is provided with available action types, general operating guidelines, the actual action taken, and its reasoning, as well as before-and-after UI states, represented by UI element representations and screenshots with SoM annotations. We request the agent to provide a concise summary of this step, including the intended action, success or failure, potential reasons for failure, and recommendations for subsequent actions. This summary will serve as the action history and be used for future action selection. See Appendix E for more details on the agent.

In addition to the full agent, we develop M3A-SIMPLE to measure the performance that can be achieved with minimal prompting, without guidelines or reflection mechanisms. This helps quantify the impact of more advanced prompting techniques and domain-specific guidance.

### 4.1.2 SEEACT BASELINE

We implement a baseline agent based on SeeAct (Zheng et al., 2024a), which was originally designed for GPT-4V for web navigation. Specifically, we implement the best-performing variant, SeeAct$_{\text{choice}}$, which grounds actions via textual choices. We implement SeeAct for the Android environment to evaluate how an existing model that performs well on web tasks (Deng et al., 2023) can be adapted and applied to Android.

To accommodate the Android environment, we adapt SeeAct in several ways. Firstly, we augment the action space from the original SeeAct implementation to support actions needed for mobile, including scroll, long press, navigate home and back, and open app actions. Secondly, in lieu of the DOM, which is not available for Android apps, we utilize the accessibility tree to construct candidate UI actions. Due to the lack of the DOM representation, we do not use the bespoke ranker model from the original implementation. However, we observe that after applying a filtering heuristic to remove non-interactable elements, the majority of screens contains less than 50 candidate elements. See Appendix E.6 for more details on the implementation.

### 4.2 EXPERIMENTAL RESULTS

We evaluate M3A, M3A-SIMPLE, and SeeAct on ANDROIDWORLD and MobileMiniWoB++. We set the seed to 30 and the temperature to 0 to aid reproducibility. Each task has a maximum allowed number of steps (detailed in Appendix F), typically set to twice the number of steps needed by human annotators to complete the task. We use Gemini 1.5 Pro, GPT-4 Turbo, and the open-source Gemma 2 27B (Team et al., 2024) as base models. For MobileMiniWoB++, we evaluate on a subset of 62 tasks, consistent with recent studies (Zheng et al., 2024c; Kim et al., 2024; Gur et al., 2022b).

Table 3 presents the success rates (SR) for the agents and human performance on both task suites. Although the agents have far from human performance, they demonstrate out-of-the-box capabilities

in operating mobile UIs, exhibiting basic understanding and control capabilities of UIs. They can perform a variety of actions, including long-press, scrolling to search for information, and revising their plan if actions do not work out. The best performance is obtained by M3A when using GPT-4. On ANDROIDWORLD the SoM-based variant is less performant, while on MobileMiniWoB++ it performs best. A similar result was obtained in recent work on computer agents for desktop applications (Xie et al., 2024). We posit SoM plays a more critical role in MobileMiniWoB++ tasks due to the often incomplete accessibility tree, compared to that of native Android apps.

The simplified agent variant M3A-SIMPLE shows a significant performance drop on ANDROID-WORLD tasks (19.8% vs 30.6% with GPT-4), indicating that additional prompting techniques and domain-specific guidance are beneficial for navigating the complexity of Android interactions. However, on MobileMiniWoB++ tasks, M3A-SIMPLE achieves comparable performance (67.7%), suggesting that these simpler tasks may not benefit as much from sophisticated prompting strategies. The open-source Gemma model's lower performance (9.5% on ANDROIDWORLD, 45.6% on MobileMiniWoB++) compared to proprietary models likely stems from its smaller parameter count, though exact comparisons are difficult as the parameter counts for GPT-4 and Gemini are not public.

### 4.3 ANALYSIS

Agents have difficulty understanding mobile UIs, often failing to detect visual cues that are essential for task completion (see Figure 6a). Additionally, agents struggle with certain UI patterns and affordances, and when they make reasoning mistakes (see Figure 6b), they often lack the capability to explore and adapt as humans do (see Figure 6c). Moreover, agents sometimes struggle with tasks that simply involve confirming system states, e.g., confirming the WiFi is turned on, suggesting challenges in both task and screen understanding.

The agents struggle with grounding, particularly when executing precise interactions, such as manipulating text (see Figure 7) or operating sliders, and they are often unable to recover from mistyping errors. In addition, for tasks that demand memory, such as performing transcriptions across apps, multiplying numbers, or scrolling, the agents struggle as they are unable to "remember" content.

SeeAct performs less effectively than M3A on the ANDROIDWORLD task suite and similarly on MobileMiniWoB++, reflecting its optimization for web rather than mobile environments. It struggles with mobile-specific actions like long-presses and swipes, and often fails to select appropriate actions due to not incorporating screen elements during action generation. Memory-intensive tasks are particularly challenging, as SeeAct only caches actions without remembering outcomes, leading to repetitive, ineffective behaviors such as endless scrolling. This lack of quick error recovery often results in task termination once maximum steps are reached.

Finally, we note that large foundation models significantly increase latency, taking three times longer than humans on average to complete tasks. On average, M3A takes 3.9 minutes to complete a task, with the text-only version taking 2.5 minutes.

## 5 ROBUSTNESS ANALYSIS

To understand agent robustness, we analyze M3A's performance across different random seeds, which generate different task parameters (e.g., calendar appointments, expense categories) and can consequently require different UI interaction patterns (e.g., scrolling to access hidden elements, handling varying numbers of elements to modify, or adapting to different input types and lengths). Across three seeds, we observe significant performance variations: 27.6%, 26.3% and 33.2% (mean 29.0%), obtained using M3A with GPT-4 Turbo with accessibility trees as input. Note that for consistency with existing literature we maintain the single-seed results in Table 3.

To better understand the sources of this variability, we evaluate agent robustness under two conditions: (1) identical tasks with the same parameters and (2) tasks with different parameter combinations, which change the initial state and task definition. We perform this analysis on a representative subset of ANDROIDWORLD tasks that span different interaction patterns and complexity levels (listed in Appendix E.4). Due to computational constraints, we conduct 20 trials for each task using our strongest agent configuration - M3A using the accessibility tree and GPT-4.

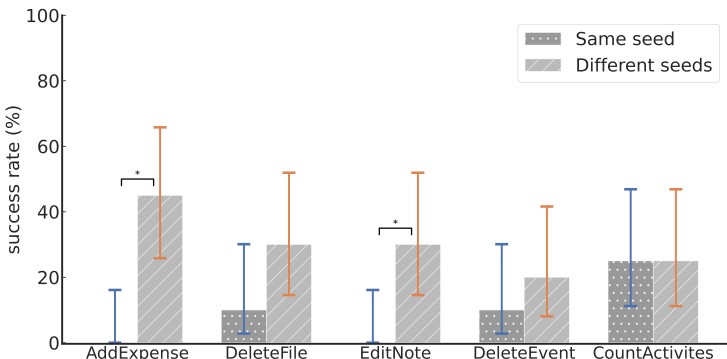

Figure 3: Success rate variation across tasks due to the parametrization built into ANDROIDWORLD. Using a fixed seed, the agent appears completely incapable of solving some tasks due to "bad luck" with the seed. In contrast, under different task parameterizations, we observe the agent can solve the tasks fairly often. Wilson binomial proportion confidence intervals (95%) are shown for the different seed group (orange) and the same seed group (blue). The different seed group has higher variance than the same seed group. Significant differences, with p-value < 0.05, are indicated by "*".

Figure 3 shows our results. With a constant seed, the agent fails on add and edit tasks and rarely solves delete tasks, primarily due to UI operation challenges. Surprisingly, performance varies even with a fixed seed, suggesting model non-determinism affects reliability. Performance varies significantly more with different seeds, with statistically significant differences for add expense and edit note tasks. The high intra-task variation indicates the model's sensitivity to task parameters. Section E.5 provides an analysis on how specific parameter variations impact agent performance.

This sensitivity aligns with observations in RL research (Henderson et al., 2018; Raffin et al., 2021; Colas et al., 2018), suggesting performance is best represented by the mean across seeds. We believe ANDROIDWORLD's support for such analysis will become increasingly valuable as more efficient models are developed. Finally, we note the observation of non-zero rewards under some seeds points to potential enhancements through RL-like mechanisms in future work.

To assess AndroidWorld's robustness to OS variations, we tested on a Pixel 5 (Android 12) alongside our primary setup (Pixel 6, Android 13). The agent achieved a 28.4% success rate, with performance variations akin to those from random seed changes, suggesting it maintained its capabilities despite differing UI layouts and device types.

These experiments underscore the importance of testing agents under varied conditions, a capability that ANDROIDWORLD effectively supports.

## 6 CONCLUSION

We introduced ANDROIDWORLD, a realistic and robust agent environment for Android that enables the development and evaluation of autonomous agents across a wide range of tasks and apps. AN-DROIDWORLD provides a reproducible task suite consisting of 116 tasks across 20 apps, with each task dynamically generated using random parameters to challenge agents with millions of unique goals. By releasing ANDROIDWORLD and establishing benchmark performance with M3A, we aim to accelerate research and development in this area, ultimately leading to the creation of computer use agents capable of operating effectively in real-world environments. Further, the dynamic nature of ANDROIDWORLD opens up new research opportunities for online learning algorithms in computer use agents.

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

## APPENDIX A  LIMITATIONS

ANDROIDWORLD currently supports open-source Android apps (>1M downloads) and built-in system apps. While testing on trending apps would be desirable, we found open-source apps often present harder challenges due to their less-optimized UIs. Popular apps typically offer more shortcuts and UI affordances, while open-source apps may require more complex interaction patterns. For example, in Figure 6c, the agent fails by repeatedly searching for a non-existent "delete-all" button instead of recognizing the need to delete notes individually.

## APPENDIX B  ETHICAL CONSIDERATIONS

**Malicious use**   There is a risk that malicious actors could engineer agents to bypass security measures like CAPTCHAs or engage in activities like spamming. Additionally, they could alter prompts or screen outputs to further harmful objectives.

**Societal impact**   Automation agents may transform societal norms, disrupt employment, and modify human behavior. While they can enhance efficiency, this improvement could pose risks if exploited by malevolent forces.

## APPENDIX C  ANDROIDWORLD ENVIRONMENT

### C.1  OBSERVATION SPACE

In ANDROIDWORLD, the Android screen is represented using a `State` class, which includes the following attributes:

- **Pixels**: An RGB array representing the current screen capture of the device. The screenshot resolution is $2400 \times 1080 \times 3$.

- **Accessibility tree**: A raw representation of the accessibility tree.[6] This UI tree provides a detailed snapshot of all UI elements currently displayed on the screen. We utilize an accessibility forwarding app from AndroidEnv (Toyama et al., 2021), which leverages gRPC to transmit the accessibility tree data efficiently to the device.

- **UI elements**: A list of processed UI elements extracted from the children of the accessibility tree. Each `UIElement` contains attributes such as text, content description, bounding boxes, and various state flags (e.g., clickable, scrollable, focused).

Since Android observations and actions are asynchronous, changes resulting from actions may take some time to manifest. Therefore, instead of using an RL-based interface, which assumes a tight coupling between actions and observations, we design an interface for the agent tailored for asynchronous interaction. This interface implements a `get_state` method responsible for capturing the current state of the environment, typically after executing an action. This method includes an optional `wait_to_stabilize` flag, which, when enabled, employs heuristics to ensure the UI elements are not in a transient state, thus providing a stable and accurate snapshot of the environment.

---

[6]Represented using all current windows; https://developer.android.com/reference/android/view/accessibility/AccessibilityWindowInfo

## C.2 ACTION SPACE

Actions are stored using a Python dataclass and executed using `adb`. The action space includes:

- Direct UI Actions:
  - Click-based actions (click, long press): Simulates touch events at specified coordinates
  - Text input: Simulates typing in focused text fields
  - Navigation: Sends home/back key events
  - Scrolling: Executes swipes in four directions (up, down, left, right)
  - App launching: Starts specified applications
- Task Management Actions:
  - Status: Reports if task is in-progress, complete, or infeasible
  - Answer: Provides responses, which are needed for information retrieval tasks
- System Actions:
  - Wait: No-op useful for loading screens and UI transitions
  - Unknown: No-op for handling internal errors

```python
ACTION_TYPES = {
    # UI Manipulation
    "CLICK": "click",
    "SCROLL": "scroll",
    "INPUT_TEXT": "input_text",
    "NAVIGATE_HOME": "navigate_home",
    "NAVIGATE_BACK": "navigate_back",
    "KEYBOARD_ENTER": "keyboard_enter",
    "OPEN_APP": "open_app",
    "LONG_PRESS": "long_press",

    # Control Flow
    "STATUS": "status",     # Reports task completion state
    "WAIT": "wait",         # Handles UI transitions
    "ANSWER": "answer",     # For information retrieval tasks
    "UNKNOWN": "unknown"    # No-op for internal errors
}

@dataclasses.dataclass()
class JSONAction:
  """Represents a parsed JSON action.

  # Example
  result_json = {'action_type': 'click', 'x': %d, 'y': %d}
  action = JSONAction(**result_json)

  Attributes:
    action_type: The action type.
    index: The index to click, if action is a click. Either an index or a <x, y>
      should be provided. See x, y attributes below.
    x: The x position to click, if the action is a click.
    y: The y position to click, if the action is a click.
    text: The text to type, if action is type.
    direction: The direction to scroll, if action is scroll.
    goal_status: If the status is a 'status' type, indicates the status of the goal.
    app_name: The app name to launch, if the action type is 'open_app'.
  """
  action_type: str
  index: int = None
  x: int = None
  y: int = None
  text: str = None
  direction: str = None
  goal_status: str = None
  app_name: str = None
```

Listing 1: Pseudo-code representation of the action space.

In addition to the UI-based action space described above, AndroidWorld provides a set of high-level APIs for direct device interaction (i.e., sending SMS messages, opening web pages, managing contacts). While the core action space focuses on fundamental UI control capabilities, these supplementary APIs found in `env/tools.py` enable future research into hybrid interaction approaches that combine both UI-based and programmatic device control.

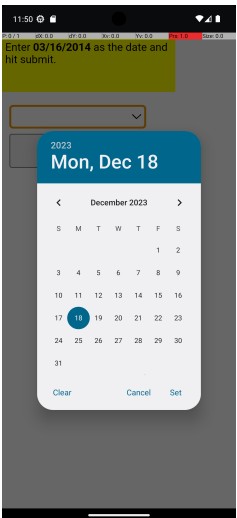

Figure 4: Native Android UI widget rendering for HTML5 <input> element.

### C.3 MOBILEMINIWOB++

Authors manually completed all tasks in MobileMiniWoB++, implemented as a WebView app, to verify solvability on a mobile interface. MobileMiniWoB++ differs from MiniWoB++ due to the touch-based interface, which required different approaches for certain tasks. For instance, highlighting text from the `highlight-text` tasks involves using Android's long-press and cursor-moving functionalities. HTML5 <input> elements are natively rendered with native Android UI widgets like the date-picker (see Figure 4).

Our implementation of MiniWoB++ contains 92 tasks in total. We exclude the following tasks: `chase-circle` (requires near-realtime movement, unachievable by humans on emulators), `moving-items` (too hard to click in emulator), `drag-cube` (drags will scroll the screen, moving the task out of view), `drag-items-grid` (elements are not interactable on Android), `drag-items` (elements are not interactable on Android), `drag-shapes` (drags will scroll the screen, moving the task out of view), `drag-sort-numbers` (elements are not interactable on Android), `text-editor` (cannot underline everything, weird glitch), `number-checkboxes` (not correctly rendered: only three columns), `use-slider-2` (slider implementation not working), `use-spinner` (slider implementation not working), and `click-menu` (the menu responsiveness breaks and the task does not behave as intended).

## APPENDIX D  ANDROIDWORLD BENCHMARK DETAILS

### D.1  APP SELECTION

Our selection of apps (summarized in Table 4) was guided by three main factors: use case, popularity, and the need for consistency and reproducibility.

**Use case and categories**  We analyzed popular app categories in app stores, focusing on productivity, communication, and multimedia. Selected apps had to meet criteria such as not requiring a login and storing data locally on the device. Additionally, we considered apps from categories that the authors commonly used, ensuring the selection was representative of real-world Android usage.

**Popularity**  We used download statistics from the Google Play Store to gauge app popularity, selecting apps with over 1 million downloads. Most of the selected apps exceeded this threshold. Less popular apps were also included if they featured common UI patterns and affordances, ensuring they are indicative of typical Android app usage. For instance, Simple Calendar Pro, though less downloaded, has a UI comparable to the widely-used Google Calendar app.

Table 4: List of ANDROIDWORLD apps and number of tasks for each one.

| App name | Description | # tasks |
|---|---|---|
| Simple Calendar Pro | A calendar app for creating, deleting, and managing events and appointments. | 17 |
| Settings | The Android system settings app for managing device settings such as Bluetooth, Wi-Fi, and brightness. | 15 |
| Markor | A note-taking app for creating, editing, deleting, and managing notes and folders. | 14 |
| Broccoli - Recipe App | A recipe management app for adding, deleting, and organizing recipes. | 13 |
| Pro Expense | An expense tracking app for adding, deleting, and managing expenses. | 9 |
| Simple SMS Messenger | An SMS app for sending, replying to, and resending text messages. | 7 |
| OpenTracks | A sport tracking app for recording and analyzing activities, durations, and distances. | 6 |
| Tasks | A task management app for tracking tasks, due dates, and priorities. | 6 |
| Clock | An app with stopwatch and timer functionality. | 4 |
| Joplin | A note-taking app. | 4 |
| Retro Music | A music player app. | 4 |
| Simple Gallery Pro | An app for viewing images. | 4 |
| Camera | An app for taking photos and videos. | 3 |
| Chrome | A web browser app. | 3 |
| Contacts | An app for managing contact information. | 3 |
| OsmAnd | A maps and navigation app with support for adding location markers, favorites, and saving tracks. | 3 |
| VLC | A media player app for playing media files. | 3 |
| Audio Recorder | An app for recording and saving audio clips. | 2 |
| Files | A file manager app for the Android filesystem, used for deleting and moving files. | 2 |
| Simple Draw Pro | A drawing app for creating and saving drawings. | 1 |

**Consistency and reproducibility** All apps were sourced from F-Droid, an open-source Android app repository. This allowed us to manage app versions precisely by selecting and distributing specific APKs. We use the newest version of each app at the time of download.

## D.2 TASK CLASSIFICATION AND GENERATION

We categorize tasks into two types: those with side-effects and those without. Tasks with side-effects are those that modify the internal state of the device or applications, such as turning off Wi-Fi or creating a calendar event. These tasks are implemented as distinct Python classes, each with its own parameter generation, initialization, evaluation, and teardown methods.

Below we show an example of the task evaluation for a `SendSms` task, which involves sending and validating a text message. The pseudocode illustrates the task initialization, success check, and parameter generation methods. Each task has its own random parameter generation method and success logic.

```python
class SendSms(TaskEval):
  """Task sending and validating a text message has been sent.

  It checks the SMS telephony database, which is located at:
  /data/data/com.android.providers.telephony/databases/mmssms.db."""

  template = (
      "Send a text message using Simple SMS Messenger to "
      "{number} with message: {message}"
  )

  def initialize_task(self, env: interface.AsyncEnv) -> None:
    """Sets up the initial state of the task."""
    super().initialize_task(env)
    clear_sms_database(env.base_env)

  def is_successful(self, env: interface.AsyncEnv) -> float:
    """Checks if the SMS was sent successfully."""
    super().is_successful(env)
    messages = get_messages(env.base_env)
    return check_message_exists(
        phone_number=self.params["number"],
        body=self.params["message"],
    )
```

```
25
26   def teardown(self, env: interface.AsyncEnv) -> None:
27     """Clears the SMS database."""
28     super().teardown(env)
29     clear_sms_database(env.base_env)
30
31   @classmethod
32   def generate_random_params(cls) -> dict[str, Any]:
33     number = generate_random_number()
34     message = generate_random_message()
35     return {
36         "number": number,
37         "message": message,
38     }
```

### D.3 INFORMATION RETRIEVAL TASKS

Tasks without side-effects are Information Retrieval tasks, requiring the agent to answer a question based on the device or app's current state. For these tasks, instead of a Python class, we create a protobuf structure to specify the prompt, parameter values, and initialization and validation logic. We decided to use a structured data format with the belief that it would allow us to define new information retrieval tasks by simply adding new entries, making it easier to scale up the number of tasks without needing to write and maintain Python classes for each one.

Initialization is defined per app, including only the state relevant to the prompt's answer and exclusion conditions for generating random states. This ensures that no random state contains information that could alter the expected answer. The initial state and prompt are parameterized using random values from the specified task parameters. For validation, we define the expected answer format within the prompt and use a few supported functions ("count", "sum", "identity") to generate the answer from the initial state.

Once an app and its specific logic are programmed, new tasks can be generated using an LLM to generate the task's protobuf. The process is not automatic and requires human review. Common issues with LLM-generated tasks include missing fields, hallucinated fields, incompatible parameter generation, insufficient parameter usage, and non-specific task prompts. We observed that the complexity of the proto structure correlates with an increase in generated task issues. Despite these challenges, we found that editing LLM-generated protobufs can be more efficient than writing a complete task from scratch.

Below we show a simplified version of the task definition for the `SimpleCalendarEventsOnDate` task which involves checking which events are on a certain date. It specifies the relevant event, the exclusion conditions for any noisy event, how to determine success, and possible parameter values to be chosen at random that will be used to fill out the task definition.

```
1  tasks {
2    name: "SimpleCalendarEventsOnDate"
3    prompt: "What events do I have {date} in Simple Calendar Pro? Answer with the titles only. If there are
         multiple titles, format your answer as a comma separated list."
4    complexity: 1
5    relevant_state {
6      // Defines information for the goal events.
7      state: {
8        calendar {
9          events {
10           start_date: "{date}"
11           start_time: "{time}"
12           duration: "{duration}"
13           title: "{title}"
14         }
15       }
16     }
17     // Non-goal events.
18     exclusion_conditions {
19       field: "start_date"
20       operation: EQUAL_TO
21       value: "{date}"
22     }
23   }
24   success_criteria {
25     expectations {
26       field_transformation {
27         operation: IDENTITY
28         field_name: "title"
29       }
30       match_type: STRING_MATCH
```

```
31        }
32     }
33
34     task_params {
35        name: "time"
36        possible_values: "11:00am"
37        // ...
38     }
39
40     task_params {
41        name: "date"
42        possible_values: "October 15 2023"
43        //...
44     }
45     task_params {
46        name: "duration"
47        possible_values: "30 m"
48        // ...
49     }
50     task_params {
51        name: "title"
52        possible_values: "Data Dive"
53        // ...
54     }
55  }
```

### D.4    HUMANS FOR TASK ANALYSIS

During development, we recruited six volunteers with proficient programming skills to analyze task difficulty, duration, and category. Each human was assigned an equal portion of tasks and tasked with identifying bugs during this annotation phase. This process resulted in the discovery and resolution of over 30 bugs.

To evaluate human performance, we enlisted two software engineers to complete the tasks using an Android emulator. Participants were provided with task descriptions and attempted to achieve the goals based on their interpretations. Each participant had one attempt per task. The majority of errors stemmed from misinterpretations or minor errors, such as entering an incorrect file extension. Other errors occurred when participants encountered unfamiliar user interfaces, impeding their ability to solve the tasks on their first attempt.

In both exercises, we informed participants about the intended use of the collected data. Participants were not required to enter any personal information in the tested tasks.

### D.5    TASK EXAMPLES

Table 5 lists some additional examples of tasks and highlights which task attributes can be parameterized in unlimited ways.

## APPENDIX E    ANDROIDWORLD AGENT DETAILS

### E.1    M3A OBSERVATIONS

ANDROIDWORLD consumes the raw screen pixels, the screen shot with Set-of-Mark (SoM) (Yang et al., 2023a) annotations, and a list of UI elements on screen.

```
1  Here is a list of descriptions for some UI elements on the current screen:
2
3  UIelement0: UIElement(text="VLC", content_description=None, class_name="android.widget.EditText",
4  bbox_pixels=BoundingBox(x_min=98, x_max=886, y_min=146, y_max=311), ...)
5  UIelement1: UIElement(text=None, content_description="Clear search box", class_name="android.widget.
       ImageButton",
6  bbox_pixels=BoundingBox(x_min=886, x_max=1023, y_min=160, y_max=297), ...)
7  UIelement2: UIElement(text="15:11", content_description="15:11", class_name="android.widget.TextView",
8  bbox_pixels=BoundingBox(x_min=50, x_max=148, y_min=1, y_max=128), ...)
9  ... More elements listed ...
10
11 ... Guidelines on action selection emitted ...
12
13 Now output an action from the above list in the correct JSON format, following the reason why you do that.
       Your answer should look like:
14
15 Reason: ...
16 Action: {"action_type":...}
```

Listing 2: The prompt format pertaining to screen representation with UI elements.

Table 5: Examples of ANDROIDWORLD tasks. We list the task nickname, the task template indicating which task attributes can be parameterized, the initialization logic that is executed before the task starts and pseudo code describing the success evaluation.

| Task nickname | Task template | Initialization logic | Success evaluation code |
|---|---|---|---|
| VlcCreatePlaylist | Create a playlist in VLC, titled "{playlist_name}" with the following files, in order: {files} | Create new mpeg files: files + "noise" files that should not be added. Add them to VLC videos folder. | `execute_sql(vlc_query) == files` |
| RecipeAddMultiple RecipesFromImage | Add the recipes from recipes.jpg in Simple Gallery Pro to the recipe app. | Write a receipt file with recipes to Simple Gallery. | `sql_rows_exist(expected_recipes)` |
| MarkorEditNote | Edit {file_name} in Markor {file_operation}. | Generate file with starting content, along with "noise" files not relevant to goal. *Note:* `file_operation` *can be to add a footer, header, or update note content.* | `file_exists(file_name, content=expected_content)` |
| ExpenseAddSingle | Add the following expenses into pro expense: {expense_csv} | Add to the app's SQLite database the expense that should be deleted, along with "noise" expenses that should not be deleted. | `sql_rows_exist(expense_obj)` |
| SimpleCalendarDelete EventsOnRelativeDay | In Simple Calendar Pro, delete all events scheduled for this {day_of_week}. | add to the app's SQLite database calendar events on specified day, along with "noise" events that should not be deleted. | `!sql_rows_exist(expected_events)` |
| FilesDeleteFile | Delete the file {file_name} from the Android filesystem located in the {subfolder} folder within the sdk_gphone_x86_64 storage area. | Generate specified file, along with "noise" files that should not be deleted. | `!file_exists(file_name)` |
| SportsTrackerActivities CountForWeek | How many {category} activities did I do this week in the OpenTracks app? Express your answer as a single integer. | add to the app's SQLite database activities for the specified category, along with "noise" activities. | `int(agent_response) == expected_count` |

## E.2 M3A ACTIONS

For the SoM prompting, the screen is annotated based on the UI elements extracted from the accessibility tree, which form the agent's action space. Figure 5 shows one example.

## E.3 ERROR ANALYSIS

We analyze M3A errors based on broader categories we observe during evaluation.

**Perceptual errors** Perceptual errors are caused when the model fails to recognize crucial elements on the screen necessary for task completion.

For the task below, the model does not recognize that the "All-day" checkbox is currently not checked (see Figure 6a):

> *In Simple Calendar Pro, create a recurring calendar event titled 'Review session for Budget Planning' starting on 2023-10-15 at 14h. The event recurs weekly, forever, and lasts for 60 minutes each occurrence. The event description should be 'We will understand software updates. Remember to confirm attendance.'*

**Reasoning errors** Reasoning errors occur when the model misinterprets the task requirements or the current state, leading to incorrect actions.

For example, in the task below, the model mistakenly believes the note name has already been entered, so it types the note text into the "Name" field and cannot recover from this error (see Figure 6b):

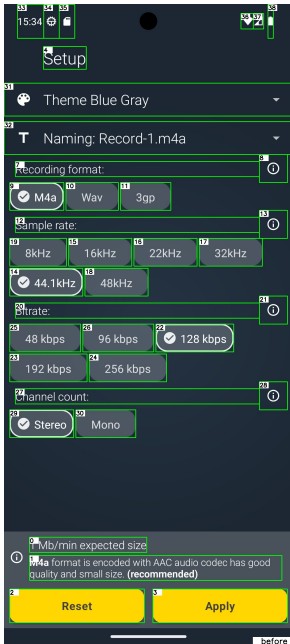

Figure 5: Set-of-marks overlaid on an Android screen.

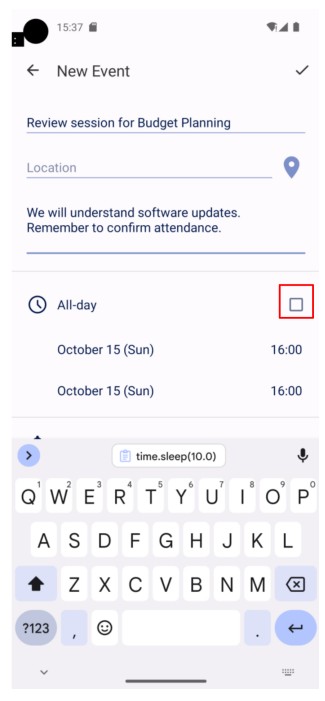 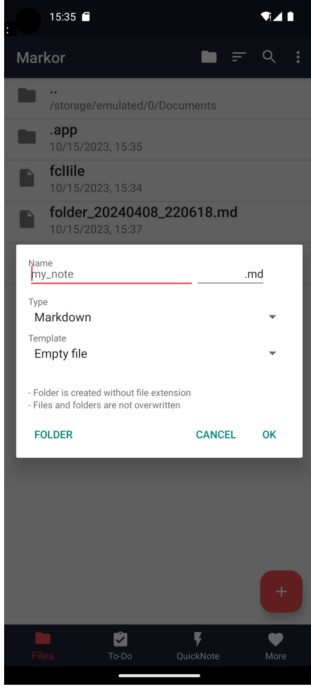 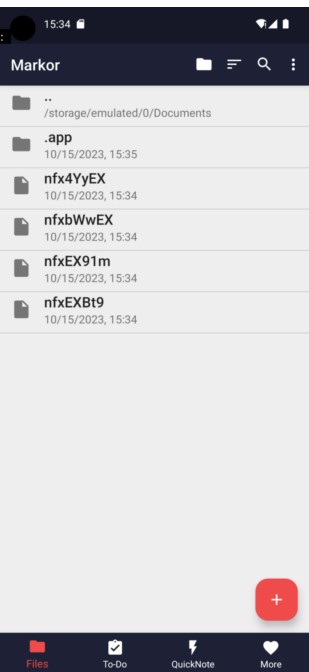

(a) Perceptual error. Red square highlights issue.

(b) Reasoning error. The agent's next action is to start entering the note's contents, which is incorrect because it needs to enter the note's name first.

(c) Missing knowledge error. To delete all notes, the agent mistakenly looks for an option to delete all the notes at once, rather than trying to do it note-by-note.

Figure 6: Screens illustrating perceptual, reasoning and missing knowledge errors.

*Create a new note in Markor named ipsum_mNcKd36pDq.txt with the following text: ...*

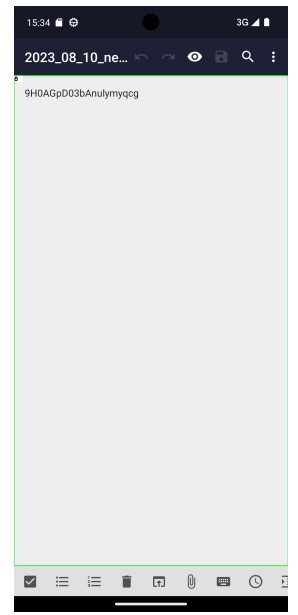
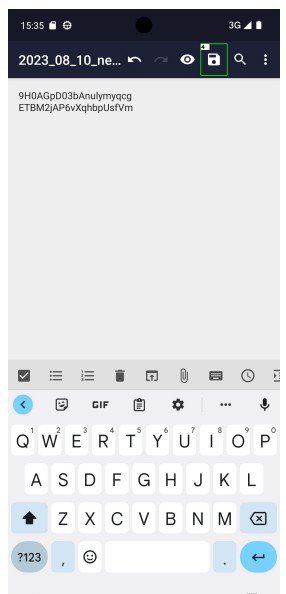

(a) Error on initial click.        (b) Error with text entered and saving.

Figure 7: Screens illustrating grounding errors.

Here is the agent's reasoning trace:

> *The user wants to create a new note with a specific name and text. The name of the note has already been entered, so all that needs to be done is to enter the text and click on the OK button.*

**Missing knowledge errors**   Missing knowledge errors occur when the model lacks the necessary understanding of the application's UI to complete a task efficiently.

For example, in the task below, the agent does not know how to delete all files at once. It looks for an option to do that instead of deleting each file one by one (see Figure 6c):

> *Delete all files in the current directory.*

**Grounding errors**   Grounding errors occur when the model fails to correctly interact with the UI elements based on their spatial or contextual positioning.

For the task below, the agent needs to update the Markor note by prepending text to the existing text. Figure 7 illustrates the errors the agent makes. It clicks the entire text field area, highlighted in green, which automatically places the cursor after the current text, resulting in the new text being appended after the current content.

> *Update the Markor note '2023_08_10_neat_wolf.txt' by adding the following text, along with a new blank line before the existing content: "ETBM2jAP6vXqhbpUsfVm", and rename it to 'sure_ocean_uRnI.txt'.*

Then, in the next screen, the text has been entered after the existing content, and the agent clicks the save button.

### E.4   AGENT ROBUSTNESS EXPERIMENTS

We ran the agent on the following tasks (the nicknames shown in the figures in parentheses):

- `MarkorEditNote` (EditNote)

- `ExpenseAddSingle` (AddExpense)
- `SimpleCalendarDeleteEventsOnRelativeDay` (DeleteEvent)
- `FilesDeleteFile` (DeleteFile)
- `SportsTrackerActivitiesCountForWeek` (CountActivities)

More details about these tasks can be found in Table 5.

## E.5 AGENT STRUGGLES DUE TO TASK PARAMETERIZATION

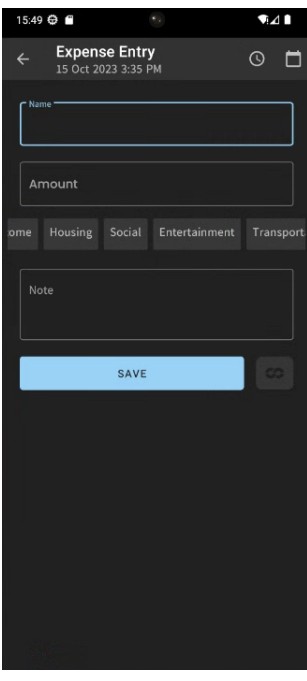

Figure 8: The expense entry interface features a horizontally scrollable category selector. When certain parameterization seeds require selecting categories that are not initially visible (e.g., "Food""), the agent fails to discover the scrolling interaction required to access them.

The variance in success rates (Figure 3) across different seeds demonstrates how task parameterization fundamentally changes task difficulty. For instance, in the `ExpenseAddSingle` task, the seed determines which expense category must be selected (see UI in Figure 8). When the seed specifies readily on-screen visible categories (e.g., "Housing", "Social"), the agent can complete the task. However, when the seed requires categories that are only accessible via horizontal scrolling (e.g., "Food", "Other"), the agent consistently fails due to its inability to discover and execute this UI interaction pattern.

Similarly, the `MarkorEditNote` task's difficulty varies based on the seed-determined variant: adding text to the top of a note, adding text to the bottom, or replacing existing text. The "replace" variant requires a more complex sequence of UI interactions (long-press, text selection, deletion, then text entry) compared to the simpler "header" variant. This explains both the complete failure under fixed seeds that happen to select challenging variants, and the higher but variable success rates when using different seeds that allow the agent to encounter various task parameterizations.

## E.6 SEEACT DETAILS

We modify the SeeAct prompt (Zheng et al., 2024a) to reflect that the environment is Android by inputting elements from the accessibility tree and supporting additional actions (e.g., scrolling). Below we include the updated prompt. We annotate the system, user, and assistant roles that are each provided to the OpenAI API.

```
1
2  > Role: SYSTEM
3  Imagine that you are imitating humans operating an Android device for a task step by step. At each stage, you
       can see the Android screen like humans by a screenshot and know the previous actions before the current
       step decided by yourself through recorded history. You need to decide on the first following action to
       take. You can tap on an element, long-press an element, swipe, input text, open an app, or use the
       keyboard enter, home, or back key. (For your understanding, they are like 'adb shell input tap', 'adb
       shell input swipe', 'adb shell input text', 'adb shell am start -n', and 'adb shell input keyevent').
       One next step means one operation within these actions. Unlike humans, for typing (e.g., in text areas,
       text boxes), you should try directly typing the input or selecting the choice, bypassing the need for an
        initial click. You should not attempt to create accounts, log in or do the final submission. Terminate
       when you deem the task complete or if it requires potentially harmful actions.
4
5  > Role: USER
6  You are asked to complete the following task: <GOAL>
7
8  Previous Actions:
9  <PREVIOUS ACTIONS>
10
11 The screenshot below shows the Android screen you see. Follow the following guidance to think step by step
       before outlining the next action step at the current stage:
12
13 (Current Screen Identification)
14 Firstly, think about what the current screen is.
15
16 (Previous Action Analysis)
17 Secondly, combined with the screenshot, analyze each step of the previous action history and their intention
       one by one. Particularly, pay more attention to the last step, which may be more related to what you
       should do now as the next step. Specifically, if the last action involved a INPUT TEXT, always evaluate
       whether it necessitates a confirmation step, because typically a single INPUT TEXT action does not make
       effect. (often, simply pressing 'Enter', assuming the default element involved in the last action,
       unless other clear elements are present for operation).
18
19 (Screenshot Details Analysis)
20 Closely examine the screenshot to check the status of every part of the screen to understand what you can
       operate with and what has been set or completed. You should closely examine the screenshot details to
       see what steps have been completed by previous actions even though you are given the textual previous
       actions. Because the textual history may not clearly and sufficiently record some effects of previous
       actions, you should closely evaluate the status of every part of the screen to understand what you have
       done.
21
22 (Next Action Based on Android screen and Analysis)
23 Then, based on your analysis, in conjunction with human phone operation habits and the logic of app design,
       decide on the following action. And clearly outline which element on the Android screen users will
       operate with as the first next target element, its detailed location, and the corresponding operation.
24
25 To be successful, it is important to follow the following rules:
26 1. You should only issue a valid action given the current observation.
27 2. You should only issue one action at a time
28 3. For handling the select dropdown elements on a screen, it's not necessary for you to provide completely
       accurate options right now. The full list of options for these elements will be supplied later.
29
30 > Role: ASSISTANT
31 <AGENT RESPONSE TO ABOVE>
32
33 > Role: USER
34 (Reiteration)
35 First, reiterate your next target element, its detailed location, and the corresponding operation.
36
37 (Multichoice Question)
38 Below is a multi-choice question, where the choices are elements on the screen. All elements are arranged in
       the order based on their height on the screen, from top to bottom (and from left to right). This
       arrangement can be used to locate them. From the screenshot, find out where and what each one is on the
       screen, taking into account both their text content and details. Then, determine whether one matches
       your target element. Please examine the choices one by one. Choose the matching one. If multiple options
        match your answer, choose the most likely one by re-examining the screenshot, the choices, and your
       further reasoning. If you would like to perform a swipe action, you can optionally select the choice
       where you will swipe.
39
40 A. "Home" icon
41 B. "Phone" icon
42 C. "Messages" icon
43 D. "Chrome" icon
44 E. "Search" icon
45 ...
46 If none of these elements match your target element, please select Z. None of the other options match the
       correct element.
47
48 (Final Answer)
49 Finally, conclude your answer using the format below. Ensure your answer is strictly adhering to the format
       provided below. Please do not leave any explanation in your answers of the final standardized format
       part, and this final part should be clear and certain. The element choice, action, and value should be
       in three separate lines.
50
51 Format:
52
53 ELEMENT: The uppercase letter of your choice. (No need for TERMINATE, KEYBOARD ENTER, WAIT, ANSWER, OPEN APP,
       NAVIGATE HOME, NAVIGATE BACK; and optional for SWIPE.)
54
55 ACTION: Choose an action from {CLICK, INPUT TEXT, LONG PRESS, NAVIGATE BACK, TERMINATE, KEYBOARD ENTER, SWIPE,
        WAIT, ANSWER, OPEN APP, NAVIGATE HOME}.
56
```

```
57  VALUE: Provide additional input based on ACTION.
58
59  The VALUE means:
60  If ACTION == INPUT TEXT, specify the text to be typed.
61  If ACTION == SWIPE, specify the direction: up, down, left, right.
62  If ACTION == OPEN APP, provide the name of the app to be opened.
63  If ACTION == ANSWER, specify the text of your answer to respond directly to a question or request for
        information.
64  For CLICK, LONG PRESS, KEYBOARD ENTER, NAVIGATE HOME, NAVIGATE BACK, WAIT, and TERMINATE, write "None".
```

## APPENDIX F    ANDROIDWORLD TASK LIST

The table below lists all tasks in ANDROIDWORLD. The maximum number of steps per task ("S") were determined based on human performance analysis, allowing agents approximately twice the number of steps typically required by human annotators to complete each task while preventing infinite loops.

Task completion tasks (e.g., send a message or edit a note) are abbreviated as "TC" and information retrieval tasks are abbreviated as "IR".

| Name | Template | Task type | Validation method | S | Apps |
|---|---|---|---|---|---|
| Audio Recorder Record Audio | Record an audio clip using Audio Recorder app and save it. | TC | Filesystem | 12 | audio recorder |
| Audio Recorder Record Audio With File Name | Record an audio clip and save it with name "{file_name}" using Audio Recorder app. | TC | Filesystem | 20 | audio recorder |
| Browser Draw | Open the file task.html in Downloads in the file manager; when prompted open it with Chrome. Then create a drawing using the three colors shown at the top and hit submit. | TC | UI-elements | 20 | files, chrome |
| Browser Maze | Open the file task.html in Downloads in the file manager; when prompted open it with Chrome. Then navigate the X to the bottom-right cell, by using the direction buttons. | TC | UI-elements | 20 | files, chrome |
| Browser Multiply | Open the file task.html in Downloads in the file manager; when prompted open it with Chrome. Then click the button 5 times, remember the numbers displayed, and enter their product in the form. | TC | UI-elements | 22 | files, chrome |
| Camera Take Photo | Take one photo. | TC | Filesystem | 10 | camera |
| Camera Take Video | Take one video. | TC | Filesystem | 10 | camera |
| Clock Stop Watch Paused Verify | Pause the stopwatch. | TC | UI-elements | 10 | clock |
| Clock Stop Watch Running | Run the stopwatch. | TC | UI-elements | 10 | clock |
| Clock Timer Entry | Create a timer with {hours} hours, {minutes} minutes, and {seconds} seconds. Do not start the timer. | TC | UI-elements | 10 | clock |
| Contacts Add Contact | Create a new contact for {name}. Their number is {number}. | TC | Database query | 12 | contacts |

| Name | Template | Task type | Validation method | S | Apps |
|------|----------|-----------|-------------------|---|------|
| Contacts New Contact Draft | Go to the new contact screen and enter the following details: First Name: {first}, Last Name: {last}, Phone: {phone}, Phone Label: {phone_label}. Do NOT hit save. | TC | UI-elements | 12 | contacts |
| Expense Add Multiple | Add the following expenses into the pro expense: {expense_list} | TC | Database query | 40 | expense |
| Expense Add Multiple From Gallery | Add the expenses from expenses.jpg in Simple gallery to pro expense. | TC | Database query | 20 | gallery, expense |
| Expense Add Multiple From Markor | Go through the transactions in my_expenses.txt in Markor. Log the reimbursable transactions in the pro expense. | TC | Database query | 30 | markor, expense |
| Expense Add Single | Add the following expenses into the pro expense: {expense_info} | TC | Database query | 12 | expense |
| Expense Delete Duplicates | Delete all but one of any expenses in pro expense that are exact duplicates, ensuring at least one instance of each unique expense remains. | TC | Database query | 12 | expense |
| Expense Delete Duplicates2 | Delete all but one of any expenses in pro expense that are exact duplicates, ensuring at least one instance of each unique expense remains. | TC | Database query | 18 | expense |
| Expense Delete Multiple | Delete the following expenses from pro expense: {expense_list}. | TC | Database query | 20 | expense |
| Expense Delete Multiple2 | Delete the following expenses from pro expense: {expense_list}. | TC | Database query | 34 | expense |
| Expense Delete Single | Delete the following expenses from pro expense: {expense_name}. | TC | Database query | 10 | expense |
| Files Delete File | Delete the file {file_name} from the Android filesystem located in the {subfolder} folder within the sdk_gphone_x86_64 storage area. | TC | Filesystem | 10 | files |
| Files Move File | Move the file {file_name} from {source_folder} within the sdk_gphone_x86_64 storage area to the {destination_folder} within the same sdk_gphone_x86_64 storage area in the Android filesystem. | TC | Filesystem | 20 | files |
| Markor Add Note Header | Update the Markor note {file_name} by adding the following text, along with a new blank line before the existing content: "{header}". | TC | Filesystem | 12 | markor |
| Markor Change Note Content | Update the content of {file_name} to "{updated_content}" in Markor. | TC | Filesystem | 12 | markor |
| Markor Create Folder | Create a new folder in Markor named {folder_name}. | TC | Filesystem | 10 | markor |
| Markor Create Note | Create a new note in Markor named {file_name} with the following text: {text} | TC | Filesystem | 16 | markor |

| Name | Template | Task type | Validation method | S | Apps |
|---|---|---|---|---|---|
| Markor Create Note And Sms | Create a new note in Markor named {file_name} with the following text: {text}. Share the entire content of the note with the phone number {number} via SMS using Simple SMS Messenger | TC | Filesystem, database query | 18 | markor, sms |
| Markor Create Note From Clipboard | Create a note in Markor named {file_name}. Perform a paste operation in the note and save the note. | TC | Filesystem | 14 | markor |
| Markor Delete All Notes | Delete all my notes in Markor. | TC | Filesystem | 14 | markor |
| Markor Delete Newest Note | Delete the newest note in Markor. | TC | Filesystem | 10 | markor |
| Markor Delete Note | Delete the note in Markor named {file_name}. | TC | Filesystem | 10 | markor |
| Markor Edit Note | Edit {file_name} in Markor. {edit_subcommand} | TC | Filesystem | 12 | markor |
| Markor Merge Notes | Merge the contents of Markor notes {file1_name}, {file2_name} and {file3_name} (in the same order) into a new Markor note named {new_file_name} and save it. Add a new line between the content of each note. | TC | Filesystem | 78 | markor |
| Markor Move Note | In Markor, move the note {file_name} from {source_folder} to {destination_folder}. | TC | Filesystem | 14 | markor |
| Markor Transcribe Receipt | Create a file in Markor, called receipt.md with the transactions from the receipt.png. Use Simple Gallery to view the receipt. Please enter transactions in csv format including the header "Date, Item, Amount". | TC | Filesystem | 18 | gallery, markor |
| Markor Transcribe Video | Transcribe the contents of video {video_name} by watching it in VLC player (located in Download) and writing the sequence of strings shown on each frame to the text file {file_name} in Markor as a comma separated list. For example, if the first frame shows the text "edna" and the second frame shows the text "pineapple", then the text file should contains only the following text: "edna, pineapple". | TC | Filesystem | 20 | markor, vlc |
| Notes Is Todo | Is the note titled '{title}' in the Joplin app marked as a todo item? Respond with either 'True' if it is a todo or 'False' if not. | IR | String match | 10 | joplin |
| Notes Meeting Attendee Count | How many attendees were present in the meeting titled '{title}' in the Joplin app? Express your answer as just a single number. | IR | String match | 10 | joplin |

| Name | Template | Task type | Validation method | S | Apps |
|---|---|---|---|---|---|
| Notes Recipe Ingredient Count | What quantity of {ingredient} do I need for the recipe '{title}' in the Joplin app? Express your answer in the format ⟨amount⟩ ⟨unit⟩ without using abbreviations. | IR | String match | 10 | joplin |
| Notes Todo Item Count | How many to-dos do I have in the '{folder}' folder in the Joplin app? Express your answer as just a single number. | IR | String match | 10 | joplin |
| Open App Task Eval | Open the {app_name} app. Clear any pop-ups that may appear by granting all permissions that are required. | TC | System API | 10 | camera, clock, contacts, settings, dialer |
| Osm And Favorite | Add a favorite location marker for {location} in the OsmAnd maps app. | TC | Filesystem | 13 | osmand |
| Osm And Marker | Add a location marker for {location} in the OsmAnd maps app. | TC | Filesystem | 20 | osmand |
| Osm And Track | Save a track with waypoints Ruggell, Liechtenstein, Bendern, Liechtenstein in the OsmAnd maps app in the same order as listed. | TC | Filesystem | 120 | osmand |
| Recipe Add Multiple Recipes | Add the following recipes into the Broccoli app: {recipe_list} | TC | Database query | 68 | recipe |
| Recipe Add Multiple Recipes From Image | Add the recipes from recipes.jpg in Simple gallery to the Broccoli recipe app. | TC | Database query | 26 | markor, recipe |
| Recipe Add Multiple Recipes From Markor | Add the recipes from recipes.txt in Markor to the Broccoli recipe app. | TC | Database query | 48 | gallery, recipe |
| Recipe Add Multiple Recipes From Markor2 | Add the recipes from recipes.txt in Markor that take 10 mins to prepare into the Broccoli recipe app. | TC | Database query | 52 | recipe |
| Recipe Add Single Recipe | Add the following recipes into the Broccoli app: {recipe_list} | TC | Database query | 24 | recipe |
| Recipe Delete Duplicate Recipes | Delete all but one of any recipes in the Broccoli app that are exact duplicates, ensuring at least one instance of each unique recipe remains | TC | Database query | 10 | recipe |
| Recipe Delete Duplicate Recipes2 | Delete all but one of any recipes in the Broccoli app that are exact duplicates, ensuring at least one instance of each unique recipe remains | TC | Database query | 24 | recipe |
| Recipe Delete Duplicate Recipes3 | Delete all but one of any recipes in the Broccoli app that are exact duplicates, ensuring at least one instance of each unique recipe remains | TC | Database query | 34 | recipe |
| Recipe Delete Multiple Recipes | Delete the following recipes from Broccoli app: {recipe_list} | TC | Database query | 24 | recipe |

| Name | Template | Task type | Validation method | S | Apps |
|---|---|---|---|---|---|
| Recipe Delete Multiple Recipes With Constraint | Delete the recipes from Broccoli app that use {ingredient} in the directions. | TC | Database query | 40 | recipe |
| Recipe Delete Multiple Recipes With Noise | Delete the following recipes from Broccoli app: {recipe_list} | TC | Database query | 34 | recipe |
| Recipe Delete Single Recipe | Delete the following recipes from Broccoli app: {recipe_list} | TC | Database query | 10 | recipe |
| Recipe Delete Single With Recipe With Noise | Delete the following recipes from Broccoli app: {recipe_list} | TC | Database query | 20 | recipe |
| Retro Create Playlist | Create a playlist in Retro Music titled "{title}" with the following songs, in order: {song_list} | TC | Database query | 24 | music |
| Retro Playing Queue | Add the following songs, in order, {song_list} to my playing queue in Retro music. | TC | Database query | 32 | music |
| Retro Playlist Duration | Create a playlist in Retro Music titled "{title}" with a duration between 45 and 50 minutes using the provided songs. | TC | Database query | 30 | music |
| Retro Save Playlist | Create a playlist in Retro Music titled "{title}" with the following songs, in order: {song_list}. Then export the playlist to the Downloads directory on the device. | TC | Database query | 50 | music |
| Save Copy Of Receipt Task Eval | Copy {file_name} in DCIM and save a copy with the same name in Download | TC | Filesystem | 16 | gallery |
| Simple Calendar Add One Event | In Simple calendar, create a calendar event on {year}-{month}-{day} at {hour}h with the title '{event_title}' and the description '{event_description}'. The event should last for {duration_mins} mins. | TC | Database query | 34 | calendar |
| Simple Calendar Add One Event In Two Weeks | In Simple calendar, create a calendar event in two weeks from today at {hour}h with the title '{event_title}' and the description '{event_description}'. The event should last for {duration_mins} mins. | TC | Database query | 20 | calendar |
| Simple Calendar Add One Event Relative Day | In Simple calendar, create a calendar event for this {day_of_week} at {hour}h with the title '{event_title}' and the description '{event_description}'. The event should last for {duration_mins} mins. | TC | Database query | 34 | calendar |
| Simple Calendar Add One Event Tomorrow | In Simple calendar, create a calendar event for tomorrow at {hour}h with the title '{event_title}' and the description '{event_description}'. The event should last for {duration_mins} mins. | TC | Database query | 26 | calendar |

| Name | Template | Task type | Validation method | S | Apps |
|---|---|---|---|---|---|
| Simple Calendar Add Repeating Event | In Simple calendar, create a recurring calendar event titled '{event_title}' starting on {year}-{month}-{day} at {hour}h. The event recurs {repeat_rule}, forever, and lasts for {duration_mins} minutes each occurrence. The event description should be '{event_description}'. | TC | Database query | 28 | calendar |
| Simple Calendar Any Events On Date | Do I have any events {date} in Simple calendar? Answer with the titles only. If there are multiples titles, format your answer in a comma separated list. | IR | Database query | 10 | calendar |
| Simple Calendar Delete Events | In Simple calendar, delete all the calendar events on {year}-{month}-{day} | TC | Database query | 14 | calendar |
| Simple Calendar Delete Events On Relative Day | In Simple calendar, delete all events scheduled for this {day_of_week}. | TC | Database query | 12 | calendar |
| Simple Calendar Delete One Event | In Simple calendar, delete the calendar event on {year}-{month}-{day} at {hour}h with the title '{event_title}' | TC | Database query | 12 | calendar |
| Simple Calendar Event On Date At Time | What is on my schedule for {date} at {time} in Simple calendar? Answer with the titles only. If there are multiples titles, format your answer in a comma separated list. | IR | Database query | 10 | calendar |
| Simple Calendar Events In Next Week | What events do I have in the next week in Simple calendar? Answer with the titles only. If there are multiples titles, format your answer in a comma separated list. | IR | Database query | 10 | calendar |
| Simple Calendar Events In Time Range | Do I have any events between {start_time} and 8pm {date} in Simple calendar? Answer with the titles only. If there are multiples titles, format your answer in a comma separated list. | IR | Database query | 10 | calendar |
| Simple Calendar Events On Date | What events do I have {date} in Simple calendar? Answer with the titles only. If there are multiple titles, format your answer as a comma separated list. | IR | Database query | 10 | calendar |
| Simple Calendar First Event After Start Time | What is my first event after {time} {date} in Simple calendar? Answer with the titles only. If there are multiples titles, format your answer in a comma separated list. | IR | Database query | 10 | calendar |
| Simple Calendar Location Of Event | What is the location of my {title} event in Simple calendar? Answer with the location only. | IR | Database query | 10 | calendar |
| Simple Calendar Next Event | What is my next upcoming event in Simple calendar? Answer with the title only. If there are multiples titles, format your answer in a comma separated list. | IR | Database query | 10 | calendar |

| Name | Template | Task type | Validation method | S | Apps |
|---|---|---|---|---|---|
| Simple Calendar Next Meeting With Person | When is my next meeting with {person} in Simple calendar? Express your answer in the format ⟨month name⟩ ⟨day⟩ ⟨year⟩ ⟨hour in 24-hour format⟩:⟨minutes⟩. | IR | Database query | 10 | calendar |
| Simple Draw Pro Create Drawing | Create a new drawing in Simple Draw Pro. Name it {file_name}. Save it in the Pictures folder within the sdk_gphone_x86_64 storage area. | TC | Filesystem | 18 | simpledrawpro |
| Simple Sms Reply | Reply to {number} with message: {message} in Simple SMS Messenger | TC | Database query | 12 | sms |
| Simple Sms Reply Most Recent | Reply to the most recent text message using Simple SMS Messenger with message: {message} | TC | Database query | 12 | sms |
| Simple Sms Resend | Resend the message I just sent to {name} in Simple SMS Messenger | TC | Database query | 12 | sms |
| Simple Sms Send | Send a text message using Simple SMS Messenger to {number} with message: {message} | TC | Database query | 12 | sms |
| Simple Sms Send Clipboard Content | Send a message to {number} with the clipboard content in Simple SMS Messenger | TC | Database query | 12 | sms |
| Simple Sms Send Received Address | Text the address of the event to {name1} that {name2} just sent me in Simple SMS Messenger | TC | Database query | 18 | sms |
| Sports Tracker Activities Count For Week | How many {category} activities did I do this week in the OpenTracks app? Express your answer as a single integer. | IR | String match | 10 | sportstracker |
| Sports Tracker Activities On Date | What activities did I do {date} in the OpenTracks app? Answer with the category only. If there are multiples categories, format your answer in a comma separated list. | IR | String match | 20 | sportstracker |
| Sports Tracker Activity Duration | How long was my {category} activity {date} in the OpenTracks app? Express your answer in minutes as a single integer. | IR | String match | 12 | sportstracker |
| Sports Tracker Longest Distance Activity | What was the longest distance covered in a {category} activity in the OpenTracks app this week? Express your answer in meters as a single integer. | IR | String match | 10 | sportstracker |
| Sports Tracker Total Distance For Category Over Interval | What was the total distance covered for {category} activities in the OpenTracks app from {start_date} to {end_date}? Express your answer in meters as a single integer. | IR | String match | 22 | sportstracker |
| Sports Tracker Total Duration For Category This Week | What was the total duration of {category} activities in the OpenTracks app this week? Express your answer in minutes as a single integer. | IR | String match | 16 | sportstracker |
| System Bluetooth Turn Off | Turn bluetooth off. | TC | System API | 10 | settings |

| Name | Template | Task type | Validation method | S | Apps |
|------|----------|-----------|-------------------|---|------|
| System Bluetooth Turn Off Verify | Turn bluetooth off. | TC | System API | 10 | settings |
| System Bluetooth Turn On | Turn bluetooth on. | TC | System API | 10 | settings |
| System Bluetooth Turn On Verify | Turn bluetooth on. | TC | System API | 10 | settings |
| System Brightness Max | Turn brightness to the max value. | TC | System API | 10 | settings |
| System Brightness Max Verify | Turn brightness to the max value. | TC | System API | 10 | settings |
| System Brightness Min | Turn brightness to the max value. | TC | System API | 10 | settings |
| System Brightness Min Verify | Turn brightness to the max value. | TC | System API | 10 | settings |
| System Copy To Clipboard | Copy the following text to the clipboard: {clipboard_content} | TC | System API | 10 | n/a |
| System Wifi Turn Off | Turn wifi off. | TC | System API | 10 | settings |
| System Wifi Turn Off Verify | Turn wifi off. | TC | System API | 10 | settings |
| System Wifi Turn On | Turn wifi on. | TC | System API | 10 | settings |
| System Wifi Turn On Verify | Turn wifi on. | TC | System API | 10 | settings |
| Tasks Completed Tasks For Date | Which tasks have I completed for {date} in Tasks app? Answer with the titles only. If there are multiples titles, format your answer in a comma separated list. | IR | String match | 10 | tasks |
| Tasks Due Next Week | How many tasks do I have due next week in Tasks app? Express your answer as a single integer. | IR | String match | 12 | tasks |
| Tasks Due On Date | What tasks do I have due {date} in Tasks app? Answer with the titles only. If there are multiples titles, format your answer in a comma separated list. | IR | String match | 10 | tasks |
| Tasks High Priority Tasks | What are my high priority tasks in Tasks app? Answer with the titles only. If there are multiples titles, format your answer in a comma separated list. | IR | String match | 10 | tasks |
| Tasks High Priority Tasks Due On Date | Which tasks with high priority are due {date} in the Tasks app? Answer with the title only. If there are multiples titles, format your answer in a comma separated list. | IR | String match | 10 | tasks |
| Tasks Incomplete Tasks On Date | What incomplete tasks do I have still have to do by {date} in Tasks app? Answer with the titles only. If there are multiples titles, format your answer in a comma separated list. | IR | String match | 10 | tasks |

| Name | Template | Task type | Validation method | S | Apps |
|------|----------|-----------|-------------------|---|------|
| Turn Off Wifi And Turn On Bluetooth | Turn off WiFi, then enable bluetooth | TC | String match | 20 | settings |
| Turn On Wifi And Open App | Turn on Wifi, then open the {app_name} app | TC | String match | 20 | settings |
| Vlc Create Playlist | Create a playlist titled {title}" with the following files in VLC (located in Internal Memory/VLCVideos), in order: {video_names} | TC | String match | 28 | vlc |
| Vlc Create Two Playlists | Create a playlist titled "{title1}" with the following files in VLC (located in Internal Memory/VLCVideos), in order: {video_names1}. And then, create a playlist titled "{title2}" with the following files in VLC, in order: {video_names2}. | TC | String match | 48 | vlc |

