# OpenReview forum: "AndroidWorld: A Dynamic Benchmarking Environment for Autonomous Agents"
_ICLR.cc/2025/Conference — ICLR 2025 Poster_

### Official Review · Reviewer_EDMT · 2024-10-27

**Soundness:** 3
**Presentation:** 3
**Contribution:** 3
**Rating:** 8
**Confidence:** 4

**Summary:**

This paper presents AndroidWorld, a realistic and reproducible Android benchmark that provides functional rewards for 116 dynamically constructed tasks across 20 real-world Android apps. The experiments show that the developed agent M3A can complete only 30.6% of the tasks, and existing web agents struggle to transfer well in Android environments.

**Strengths:**

1. The paper is overall clear and well-written.
2. The environment is lightweight and compatible with task random parameterization.
3. Detailed examples and error analysis are provided in the appendix.
4. I appreciate the effort to implement the initialization and teardown process for each task to ensure reliability and reproducibility.

**Weaknesses:**

1. The experimental setup is not very clear. Specifically:
- What specific stop criteria are used for the experiments (e.g., step limits or time limits)? How is the task-specific step budget set for each task?
- What decoding temperature is applied?
- How is human performance measured?

2. It would be helpful to include results from more models, including recent APIs and open-source models. The ablation study only investigates the observation space and does not consider the impact of other agent design choices, such as Reflexion.

3. Are all tasks and apps offline? If not, how to ensure reproducibility? For example:
- Does the SendSms task send actual messages?
- Could real-world time and date affect the evaluation results and reproducibility in calendar/timer tasks within AndroidWorld and MobileMiniWoB++?

4. More discussion of related work, e.g.:
- Zheng et al., "Agentstudio: A toolkit for building general virtual agents," arXiv preprint arXiv:2403.17918.
- You et al., "Ferret-UI: Grounded Mobile UI Understanding with Multimodal LLMs." arXiv preprint arXiv:2404.05719.

**Questions:**

1. What is the number of human hours required to develop the environment and configure the tasks?
2. In your initialization and teardown process, have you considered scenarios where the agent performs actions outside the defined teardown process, potentially altering system states and resulting in non-identical states across different experiment trials?

---

> ### Author Response · Authors · 2024-11-22
>
> We thank the reviewer for their review and helpful comments. We address them one-by-one:
>
> **W1.** We are happy to provide these details about the experimental setup and have updated the manuscript accordingly (see Section 4.2). Thank you for helping us improve on the clarity of our presentation.
>
> a) Stop criteria: Task-specific maximum steps are determined based on the human step count analysis shown in Figure 2b, typically set to 2x the measured human steps. This accounts for agent exploration while preventing infinite loops. For even more clarity, we’ve added Appendix E listing out details of each task, including the maximum number of steps.
>
> b) Decoding temperature: We use temperature=0 for both GPT-4 and Gemini to aid reproducibility.
>
> c) As described in Appendix C.4, to obtain human performance, we recruited two software engineers who completed all tasks using an Android emulator. They operated the emulator using the keyboard and mouse. Each participant had one attempt per task, and their results were averaged. The majority of errors stemmed from interface unfamiliarity or minor misinterpretations rather than fundamental task difficulty.
>
> **W2.** Thank you for this suggestion. We have expanded our evaluation by adding open-source baselines and agent variants. These updates are reflected in the manuscript (Section 4.2). We evaluated Gemma 2 (27B parameters) as an accessible baseline, achieving a 9.5% success rate on AndroidWorld. In addition to the original M3A agent, we introduce M3A-Simple, a baseline agent without reflection mechanisms or task-specific guidelines, achieving a 19.8% success rate (compared to M3A's 30.6%). These additional evaluations help researchers understand both model and architectural contributions to performance.
>
> **W3.** For safety reasons, all tasks operate entirely offline within a controlled emulator environment. For SMS-related tasks, we leverage the emulator's SMS simulation capabilities - while the UI shows normal messaging behavior, no actual messages are transmitted outside the emulator. The messaging state is verified through the emulator's messaging database.
> Regarding how real-world time and date affect the evaluation results and reproducibility, this is an important point. As noted in Section 3.3, we address this by setting a fixed system time (specifically October 15th, 2023 15:34:00:0000) to maintain reproducibility of time-sensitive tasks like calendar events and timers. We further clarify in the manuscript that we set the date time during the task initialization across all task executions.
>
> **W4.** We have added the two papers to the Related Work section. Below we describe their relevance in more detail:
>
> AgentStudio is a toolkit for developing and evaluating generalist virtual agents, offering interactive environments, benchmarks, and datasets. It supports both GUI and API actions. The suite includes 205 tasks across multiple apps and provides additional datasets (GroundUI, IDMBench, CriticBench) for GUI grounding, action prediction, and success detection. Task completion is determined as in OSWorld. However, AgentStudio is limited to desktop applications and does not support mobile environments, leaving a critical gap in benchmarking agents for mobile platforms. In addition, unlike AndroidWorld, AgentStudio does not support unlimited task parameterizations which makes robustness analysis infeasible. The benchmark is also limited to 9 apps.
>
> Ferret-UI adapts the Ferret multimodal LLM for UI understanding by incorporating 'any resolution' processing to handle mobile screens' aspect ratios and small UI elements, demonstrating strong performance on UI referencing and grounding tasks compared to GPT-4V and other baselines. The focus of Ferret-UI is focused on building a generalist model for user interface (UI) understanding, a critical capability of UI control agents, but not on a benchmark for multi-step task evaluation.

---

> > ### Author Response · Authors · 2024-11-22
> >
> > **Q1.** The development of AndroidWorld spanned approximately 10 months of development by a group of 5+ people. However, this number doesn't tell the complete story. A significant portion of the effort went into building abstractions and reusable components for later task additions. For example:
> > * Environment interaction interfaces and utilities for device interaction
> > * Generic database exploration and validation logic shared across multiple apps
> > * File system operation abstractions
> > * Task parameterization infrastructure
> >
> > Moreover, to make the framework usable and fast, lots of work went into debugging and improving the implementation, based on feedback provided by early adopters.
> >
> > Using this infrastructure, onboarding a new app and implementing several tasks for it typically takes anywhere from a few hours to a couple of days, depending on the app's complexity and state management requirements.
> >
> > **Q2.** Thank you for the question – this is an important observation about potential state contamination. While agents could theoretically modify global system states outside our initiization/teardown process, we made a conscious engineering tradeoff here. We could ensure perfect isolation by loading a fresh device snapshot for each task, but this would significantly increase evaluation latency having to load it each time. In practice, we haven't observed agents modifying system settings outside their task scope - they typically operate within the presented UI context. Thus we believe our task-specific initialization/teardown represents a good balance between absolute reproducibility and evaluation efficiency.

---

> > > ### Comment · Reviewer_EDMT · 2024-11-24
> > >
> > > Thanks for addressing my questions. I have increased my score.

---

### Official Review · Reviewer_5yqj · 2024-10-30

**Soundness:** 3
**Presentation:** 3
**Contribution:** 3
**Rating:** 6
**Confidence:** 4

**Summary:**

This paper introduces AndroidWorld, an executable, reproducible, and realistic benchmark for evaluating Android device control agents. The authors also present results from several baseline agents and human.

**Strengths:**

1. Solid contribution: this benchmarks fills the gap for solid, reproducible, and executable benchmarks for Android device control tasks.
2. Good presentation: the presentation is clear, discussions in related works are comprehensive
3. Interesting experiments: the experiments presents a few good baselines and shows how human performs on the tasks. The robustness analysis also provides new information to the community.

**Weaknesses:**

1. Lack of many real-world apps and tasks: The benchmark lacks many real-world applications and tasks, as ensuring full reproducibility and automated evaluation makes it impossible to include closed-source apps like YouTube, Twitter (X), Amazon, or actual real-web browsing. This results in inherent sim-to-real gaps.
2. Lack of device diversity on android device/OS: The emulated device is fixed as a Pixel 6 running Android 13. However, in practice, what we care about is agents' performance across various devices and OS versions. Adding more OS/device options would potentially make the benchmark more correlated with real world use cases.
3. Lack of open-source baselines: All baselines in the paper appear to be based on proprietary models. Including some open-source models as baselines would be beneficial, as it would provide researchers a baseline they can iterate on.

**Questions:**

1. How was the confidence interval in Figure 3 calculated? I couldn’t find details on how these intervals were derived. Did you run the same model on these tasks multiple times to obtain them?

---

> ### Author Response · Authors · 2024-11-22
>
> We thank the reviewer for their valuable comments and suggestions. We address each of them one-by-one.
>
> **W1.** Creating benchmarks inherently involves balancing accessibility and reproducibility. Building reproducible automation tasks using closed-source apps that sync with backend services is next-to-impossible because the benchmark is unable to control state on these services. Other major benchmarks have addressed this challenge in similar ways:
> 1. WebArena uses open-source website alternatives with static data like Postmill instead of Reddit and OneStopMarket instead of Amazon.
> 1. In order to prevent copyright issues, OSWorld primarily relies on FOSS applications like LibreOffice and GIMP.
> 1. Even attempts to include real-world services (like OSWorld's TripAdvisor tasks) face reproducibility issues due to changing backend data.
>
> AndroidWorld makes this same necessary tradeoff while providing three key advances:
>
> First, as the first comprehensive, reproducible mobile benchmark, our environment spans 20 diverse apps covering mobile-specific interaction patterns fundamentally different from desktop environments.
>
> Second, our parameterization system generates unlimited task variations, providing richer evaluation than static tasks - our analysis (Section 5) shows even state-of-the-art models vary significantly (26.3% to 33.2%) across different parameterizations.
>
> Third, we provide robust state management tools that guarantee consistent evaluation conditions while supporting this rich parameterization. The UI patterns in our selected apps parallel commercial apps and often present more challenging scenarios by lacking heavily optimized UIs, making them well-suited for evaluating fundamental agent capabilities.
>
> **W2.** We strongly agree about the importance of device/OS diversity. To validate AndroidWorld's cross-version capabilities, we successfully ran the benchmark on a different emulator (Pixel 5) and Android 12. We have updated the manuscript accordingly to address that AndroidWorld is also cross-platform as well. Please see results in Section 5.
>
> However, we still feel it is important to standardize one (e.g., Pixel 6 (API 33)) as the primary evaluation environment to ensure reproducible comparisons between different approaches and future work. Researchers can use our cross-version validation results to understand potential performance variations while having a stable reference point for comparing different approaches.
>
> **W3.** We appreciate this suggestion. We have expanded our evaluation by adding open-source baselines and agent variants. We evaluated Gemma 2 (27B parameters) achieving 9.5% success rate. In addition to the original M3A agent, we introduce M3A-Simple, a baseline agent without reflection mechanisms or task-specific guidelines, and we run Gemma 2 on it achieving 3.4%. These new evaluations help researchers understand both model and architectural contributions to performance. These updates are reflected in the manuscript in Section 4.2 and Table 3.
>
> **Q1.** Thank you for this important question about our statistical methodology. As noted in the Figure 3 caption, we used Wilson binomial proportion confidence intervals at 95% confidence level for our analysis. Our choice of Wilson intervals was deliberate, given their suitability for binary outcomes (success/failure), strong performance with small sample sizes (N=20 in our experiments), and appropriate handling of boundary conditions. These intervals provide reliable coverage even for extreme success rates (0 or 1), making them particularly appropriate for our experimental setup.
>
> Section 4.4 details our experimental setup, where we conducted 20 trials per task under each condition (fixed seed vs different seeds) to ensure robust statistical analysis. We appreciate the opportunity to clarify these methodological details.

---

> ### Comment · Reviewer_5yqj · 2024-11-22
>
> Given that authors have addressed W2 and W3, I'm keeping my positive score.
> In general, having a reproducible environment on Android device control is a great contribution to the community.

---

> > ### Author Response · Authors · 2024-11-22
> >
> > Thank you for your positive feedback. We noticed your initial comment (notified via email) indicated an intention to increase the score to a 7 based on our addressing W2 and W3. We just wanted to confirm if this was still your intention, and if there are any remaining concerns we haven't fully addressed.

---

### Official Review · Reviewer_gUrJ · 2024-11-04

**Soundness:** 4
**Presentation:** 3
**Contribution:** 4
**Rating:** 8
**Confidence:** 4

**Summary:**

A benchmark for LLM (or LMM) agents controlling mobile devices is introduced, focusing on the necessity of interactive evaluation (as common benchmarks in this domain have mainly focused on static datasets so far) and the agents’ generalization ability in terms of task instructions. The tasks regard applications in Android commonly used in daily life as well as tasks in MiniWob++, which have been a common benchmark in digital device control. The study also proposes a new prompting approach to enhance agentic capabilities in foundation models and presents a detailed analysis of agent performance. The analyses include comparisons across different input modalities, robustness across varied task parameters, and their common errors.

The main contributions of this work are the introduction of a new benchmark, tasks that can be randomized via instruction parametrization, and a new algorithm enhancing the capability of LLM agents for mobile device control.

**Strengths:**

S1 - This work presents a foundational contribution that advances the AI community's understanding of **mobile device control**, offering a robust framework for evaluating LLM agents in interactive environments.

S2 - The related work and comparisons to existing benchmarks are well-organized and comprehensive. This provides a clear context of how this benchmark builds upon and differentiates itself from prior studies.

S3 - The benchmark introduces an important challenge related to task generalization by implementing parametrized tasks. This effectively assesses the generalizability of agents across varying task instructions.

**Weaknesses:**

W1 - The explanation of the action space could be more detailed. For instance, in the ACTION_TYPE section described in Appendix B.2, further clarification on the purpose of the "STATUS" action would be helpful. Additionally, what is the rationale behind the necessity of a "SWIPE" action, despite the existence of "SCROLL", could be further justified.

W2 - Although line 257 mentions a limited set of high-level APIs, Appendix B lacks specific details on these APIs. More explanation would help us understand the scope of actions.

W3 - The rationale behind the task categorization tags could be elaborated on further. Expanding on the justification for the categorization approach would clarify its significance and enhance the robustness of the framework.

W4 - I believe that this work could benefit from the inclusion of more baseline comparisons. Presenting results from a basic version of the agent, without any prompting methods for example, would provide valuable reference points. While the current results demonstrate that M3A is a strong baseline, additional baselines would enrich the study and offer further insights into performance.

**Questions:**

Q1 - Could the authors explain the rationale for integrating MiniWoB++ into a mobile environment, as this incorporation seems highly auxiliary? What unique challenges do these MiniWob++ tasks present, especially compared to tasks involving typical applications in AndroidWorld?

Q2 - While the unique aspects of the mobile action space are described, not all are represented in the current action space (e.g., multi-finger actions). Could the authors comment on the extensibility of M3A/AndroidWorld to accommodate different action spaces, or do they plan to incorporate a broader range of actions in the future?

Q3 - Could the authors provide a guideline on the training and testing configuration, such as a split of task categories? Additional details would aid in understanding the optimal configuration for achieving reliable results.

Q4 - (Minor) Would the authors consider providing results using Claude’s newly proposed models (considering that the relevant developers are interested in digital device usage)? This addition would offer a timely perspective on how recent advancements perform within this framework.

Q5 - The authors mention that using multiple random seeds and averaging the results would provide a fair representation of the results (line 493). Could the authors confirm if the results in Table 3 are also based on averaged results across different seeds? If not, might it be beneficial to conduct the experiments across multiple seeds, as task randomness can significantly impact agent performance?

---

> ### Author Response · Authors · 2024-11-22
>
> We thank the reviewer for their helpful comments. We address them one-by-one.
>
> **W1.** Thank you for your careful review and catching this discrepancy.
> 1. The SCROLL action is sufficient for all navigation needs in our current tasks. Our actual implementation only uses a SCROLL action. We apologize for the mistake. We corrected this in Appendix B.2 to reflect that agents only use SCROLL actions, not SWIPE.
> 1. We have added more detailed descriptions of each action type in Appendix B.2
> 1. We have clarified the purpose and usage of the STATUS action which agents use to report their understanding of task progress (i.e. in-progress, complete, or infeasible).
>
> **W2.** We appreciate the reviewer's comment about API documentation. We have expanded Appendix B.2 to include details about AndroidWorld's high-level APIs, which complement the UI-based action space. These APIs (including SMS messaging, web navigation, and contact management) provide additional capabilities for future research into hybrid interaction approaches.
>
> **W3.** Our task categorization approach used a predefined set of tags (TranscriptionSearch, Math/Counting, Game Playing, Verification, Memorization, Screen Reading, Complex UI Understanding, Repetition, Multi-app, Data Entry, Data Edit, Requires Setup, Information Retrieval, and Parameterized) developed through an initial task analysis. We deliberately chose this fixed set to:
> 1. Ensure sufficient task coverage across categories
> 1. Streamline the annotation process for human raters
> 1. Enable systematic analysis of agent performance across different interaction types. This structured approach helps identify which types of interactions are most challenging for current agents.
>
> **W4.** We thank the reviewer for this suggestion. We have now addressed this by adding new baseline comparisons in Table 3 in Section 4.2. Specifically, we evaluated:
> M3A-Simple, a baseline agent with minimal prompting – without reflection mechanisms or task-specific guidelines – which achieved a 19.8% success rate on AndroidWorld (vs M3A's 30.6%), demonstrating the value of advanced prompting techniques for complex Android tasks. Interestingly, on the MiniWoB++ tasks, M3A-Simple matched M3A's performance (67.7%), suggesting simpler tasks such as those of MiniWob++ may not benefit from the additional techniques.
> Gemma 2, an open-source model baseline, which achieved 9.5% on AndroidWorld and 45.6% on MiniWoB++, providing an accessible reference point for the research community to build upon. We also benchmark Gemma 2 with M3A-Simple, which achieved 3.4% on AndroidWorld and 35.6% on MiniWoB++.

---

> ### Author Response · Authors · 2024-11-22
>
> **Q1.** The MiniWoB++ integration serves multiple purposes:
> 1. It enables direct comparisons between mobile and web interaction patterns as well as testing on UI widgets that, although simplified and contained in very small screens, are indicative of logical flows that exist in mobile apps too. For instance, MiniWob++ includes interactions with piecharts, geometric figures, grids of various types, color palettes, etc.
> 1. While recent work like Synapse (https://arxiv.org/abs/2306.07863) has achieved super-human performance using few-shot learning and demonstrations, zero-shot performance remains challenging on MiniWob++.
> 1. Having MiniWoB++ in the same platform as AndroidWorld reduces friction for researchers - they can evaluate both mobile and web interactions using a single framework.
> 1. It provides continuity with existing research while expanding into mobile automation
>
> **Q2.** Adding multi-touch support is straightforward due to our lightweight ADB wrappers. Our modular architecture makes it easy to extend the action space while maintaining backward compatibility. We have not added these actions yet because the task suite we support does not require them yet.
>
> **Q3.** While we use zero-shot evaluation (i.e., no training split), we recommend:
> 1. Using seed=30 for reproducible comparisons.
> 1. Ideally, averaging results across multiple seeds for more robust evaluation. This ensures fair comparisons while acknowledging the impact of randomization on performance.
>
> **Q4.** We appreciate this suggestion. While testing Claude's new models would be interesting, our focus in this paper was on establishing the benchmark methodology and providing initial baselines. However, we will make AndroidWorld fully accessible to the community so that others can leverage our framework to evaluate the expanding landscape of models, including Claude and others.
>
> **Q5.** Thank you for this suggestion. Given our compute budget, the results in Table 3 are for a single seed.
> However, to validate these findings at scale, we have updated the manuscript with an evaluation of our best model (M3A with GPT-4 using accessibility trees) with 3 seeds across all tasks. The results show a mean performance of 29.0% across three runs/seeds (27.6%, 26.3%, and 33.3%). Please see Section 5 for details. Notably, this differs from our single-seed result of 30.6%, which we report in Table 3 for consistency with existing literature. This variance between single-seed and averaged results reinforces our paper's emphasis on the importance of evaluating across multiple seeds for robust performance measurement. While comprehensive seed analysis for all model variants would require substantial computational resources, we believe AndroidWorld's support for such analysis will become increasingly valuable as more efficient models are developed.

---

> > ### Comment · Reviewer_gUrJ · 2024-11-25
> >
> > Thanks for clarifying and addressing many of my confusions. I keep my score, as my strong belief in the value of this work remains positive. I hope the work has become more solid by equipping more explanations and baselines.

---

### Official Review · Reviewer_1Fiv · 2024-11-04

**Soundness:** 3
**Presentation:** 2
**Contribution:** 3
**Rating:** 6
**Confidence:** 4

**Summary:**

The paper introduces AndroidWorld, a benchmarking environment for autonomous agents interacting with Android devices. This environment enables testing on various apps by dynamically constructing parameterized tasks expressed in natural language. AndroidWorld supports a more realistic and varied testing experience than static environments, enabling agents to engage with numerous unique task goals and environmental conditions. Initial experiments using baseline agents indicate significant room for improvement in task completion rates, with top-performing agents achieving a 30.6% success rate. This highlights the challenges and research opportunities in building robust, cross-platform autonomous agents.

**Strengths:**

- AndroidWorld’s dynamic task construction introduces extensive variability in task conditions, offering a realistic and reproducible environment for testing autonomous agents on Android.
- The paper provides a robust baseline evaluation and a thorough performance analysis across real-world conditions.
- Extensive experiments provide essential insights into current agents' limitations and suggest potential pathways for improvement in future cross-platform agent designs.

**Weaknesses:**

- The agents achieve a low overall success rate (30.6%), which, while reflecting the environment’s complexity, suggests that current methods may need significant refinement to handle mobile platforms effectively.
- Though this paper introduces extensive task parameterization, it does not provide a detailed explanation of how specific parameter variations impact agent performance. There is also no analysis of performance changes across different task components or parameters.
- Large foundation models integrated into the agents exhibit high latency, taking significantly longer than humans to complete tasks, which may limit the practical applicability of these agents in real-time scenarios.

**Questions:**

- How might AndroidWorld be expanded or adapted to include tasks that require specific contextual understanding, such as personalized interactions with app data?
- How might agent performance improve if AndroidWorld incorporated multimodal feedback mechanisms, such as sound or haptic responses, beyond the visual and text-based cues currently used?
- B-MoCA, like AndroidWorld, operates in Android environments and also employs various randomization schemes for task variability. How does your approach to task diversity, randomization, and reproducibility compare to B-MoCA? What unique advantages does AndroidWorld offer regarding its benchmarking scope, real-world applicability, and parameterized task generation? Additionally, are there specific aspects of your design that provide it with a distinct edge in evaluating agent robustness?
- Could the authors provide a full list of the tasks included in AndroidWorld, along with the initialization logic and success evaluation codes for each task?

---

> ### Author Response · Authors · 2024-11-22
>
> We thank the reviewer for their helpful comments and questions. Below we address them one-by-one.
>
> **W1.** The 30.6% success rate using state-of-the-art foundation models is actually the reason why this benchmark is extremely valuable. Such low performance reveals the significant challenges in mobile automation that were previously unmeasured and offers an opportunity to measure progress in this field and hill climb. Like early benchmarks in computer vision and natural language processing that showed large gaps between human and machine performance, AndroidWorld reveals important limitations in current approaches. This gap motivates further research, therefore we consider it a strength, not a weakness, of our benchmark.
>
> **W2.** While Section 5 and Figure 3 of our paper provide detailed analysis of how parameter variations affect agent performance, we appreciate this suggestion for even more comprehensive analysis. We have added section D.5 that provides an analysis on how specific parameter variations impact agent performance. Additionally, we have conducted an evaluation of our best model (M3A with GPT-4 Turbo using accessibility trees) across all tasks with three different seeds, each generating different task parameters. This evaluation shows a mean performance of 29.0% (individual runs: 27.6%, 26.3%, and 33.2%), reinforcing our finding that task parameterization significantly impacts agent success rates. We added this result at the beginning of Section 5. This additional analysis strengthens our understanding of how agents perform under different task conditions.
>
> **W3.** The latency observation is an artifact of current LLM technology, not a limitation of AndroidWorld itself. Our benchmark is model-agnostic and can evaluate any agent implementation. This is why our evaluation includes different base models (GPT-4,Gemini, Gemma) and agent variations.  Our paper’s main contribution is  providing a comprehensive evaluation framework - execution speed is a separate concern that can be addressed through model optimization, distillation, or alternative architectures. We also would like to point out that latency is an issue affecting most LLM-based applications and that UI control agents are becoming a driving use case of LLM technology, with related announcements from companies leading in this space (Google, Anthropic, OpenAI).

---

> ### Author Response · Authors · 2024-11-22
>
> **Q1.** AndroidWorld's architecture readily supports contextual understanding through its state management system (e.g., AndroidWorld can capture location information). We could extend this for personalized interactions by:
>
> * Creating distinct user personas (e.g., student, professional, parent) with associated behavioral patterns, history, and data footprints.
> * Injecting persona-specific data into app databases and shared preferences during task initialization (e.g., calendar events matching a student's class schedule).
> * Adding contextual parameters to our task generation system (time of day, location, device settings).
> * Supporting cross-app context (e.g., tasks that reference data from both calendar and messaging apps).
>
> For example, a 'student' persona might include class schedules in calendar, academic notes in Markor, and study-related reminders in Tasks. A 'professional' persona might have work meetings, expense reports, and business contacts. Our state management framework ensures these extensions maintain reproducibility while enabling more realistic evaluation scenarios.
>
> The code structure in Section C.2 demonstrates how this could be implemented, and our open architecture allows researchers to contribute additional personas and contexts.
>
> **Q2.** While multimodal feedback is definitely a very interesting direction, the current scope deliberately matches real-world Android interaction patterns. Most Android apps primarily rely on visual and touch interactions, making these the most critical modalities to benchmark.
>
> **Q3.** There are crucial differences between AndroidWorld's and B-MoCA's approaches to reproducibility and task generation.
>
> Reproducibility and privacy: While B-MoCA relies on personal accounts and login processes that create significant reproducibility challenges (as documented in their codebase: 'using the personal account might cause a leak of privacy information'), AndroidWorld is designed for complete reproducibility with no external dependencies. We deliberately selected apps that store data locally and don't require authentication, ensuring that any researcher can reproduce our exact task conditions.
> Furthermore, B-MoCA's approach introduces security and privacy risks by requiring researchers to provide account credentials in configuration files. In contrast, AndroidWorld's state management system uses Android's native mechanisms to create and validate task states, making it both more reliable and more secure. Our approach ensures consistent behavior across different research environments without compromising privacy or reproducibility.
> This architectural difference is not merely an implementation detail - it fundamentally affects the benchmark's utility for the research community. While B-MoCA's tasks may superficially appear similar, the dependence on external accounts and services makes it impossible to guarantee consistent evaluation conditions across different research groups or time periods.
>
> Success detection: B-MoCA provides rule-based success detectors specific to each task instance, and therefore it does *not* support automated task parameterization at scale. Most of B-MoCA’s success detectors are implemented by inspecting the value of target UI elements on the screen, which is generally less reliable. For example, in a Calculator task the detector locates the “Formula” input field using a static identifier and verifies that it matches the target formula (e.g., “1+1”). All 19 tasks in the Calculator category adopt the same logic. In Call tasks, the detector inspects the “number” text field in the UI and verifies that it matches the target phone number. All 12 call tasks follow this logic. AndroidWorld makes reward signals durable across task parameterizations allowing it to support an unlimited number of task configurations. The number of task instances supported by AndroidWorld is therefore several orders of magnitude higher.
>
> **Q4.** We thank the reviewer for this suggestion. We have updated the manuscript to include a comprehensive task list along with initialization logic and evaluation codes in the Appendix E of the manuscript.

---

> > ### Comment · Reviewer_1Fiv · 2024-11-24
> >
> > Thanks for your effort and it resolves most of my concerns. I have increased my score.

---

### Meta-Review · Area_Chair_VyGU · 2024-12-19

**Metareview:**

The reviewers unanimously agree that AndroidWorld provides a novel, challenging benchmark for the computer-use abilities of LLMs. Importantly, reviewers point out that this work also reports the various aspects of these tasks with which modern LLMs struggle, providing a clear indication of what improvement on this benchmark means for LLMs capabilities.

**Additional Comments On Reviewer Discussion:**

Most reviewers agreed this work provides a significant contribution. Reviewer 1Fiv asked for additional details on the benchmark, including a comparison to B-MoCA and details on how the different environment parameters impact the difficulty of the benchmark. The authors addressed these concerns in their rebuttal.

---

### Decision · Program_Chairs · 2025-01-22

Accept (Poster)